# Mixed Robust/Average Submodular Partitioning: Fast Algorithms, Guarantees, and Applications

Kai Wei[1]      Rishabh Iyer[1]      Shengjie Wang[2]      Wenruo Bai[1]      Jeff Bilmes[1]

[1] Department of Electrical Engineering, University of Washington
[2] Department of Computer Science, University of Washington
{kaiwei, rkiyer, wangsj, wrbai, bilmes}@u.washington.edu

## Abstract

We investigate two novel mixed robust/average-case submodular data partitioning problems that we collectively call *Submodular Partitioning*. These problems generalize purely robust instances of the problem, namely *max-min submodular fair allocation* (SFA) [12] and *min-max submodular load balancing* (SLB) [25], and also average-case instances, that is the *submodular welfare problem* (SWP) [26] and *submodular multiway partition* (SMP) [5]. While the robust versions have been studied in the theory community [11, 12, 16, 25, 26], existing work has focused on tight approximation guarantees, and the resultant algorithms are not generally scalable to large real-world applications. This is in contrast to the average case, where most of the algorithms are scalable. In the present paper, we bridge this gap, by proposing several new algorithms (including greedy, majorization-minimization, minorization-maximization, and relaxation algorithms) that not only scale to large datasets but that also achieve theoretical approximation guarantees comparable to the state-of-the-art. We moreover provide new scalable algorithms that apply to additive combinations of the robust and average-case objectives. We show that these problems have many applications in machine learning (ML), including data partitioning and load balancing for distributed ML, data clustering, and image segmentation. We empirically demonstrate the efficacy of our algorithms on real-world problems involving data partitioning for distributed optimization (of convex and deep neural network objectives), and also purely unsupervised image segmentation.

## 1   Introduction

The problem of data partitioning is of great importance to many machine learning (ML) and data science applications as is evidenced by the wealth of clustering procedures that have been and continue to be developed and used. Most data partitioning problems are based on expected, or average-case, utility objectives where the goal is to optimize a sum of cluster costs, and this includes the ubiquitous $k$-means procedure [1]. Other algorithms are based on robust objective functions [10], where the goal is to optimize the worst-case cluster cost. Such robust algorithms are particularly important in mission critical applications, such as parallel and distributed computing, where one single poor partition block can significantly slow down an entire parallel machine (as all compute nodes might need to spin while waiting for a slow node to complete a round of computation). Taking a weighted combination of both robust and average case objective functions allows one to balance between optimizing worst-case and overall performance. We are unaware, however, of any previous work that allows for a mixing between worst- and average-case objectives in the context of data partitioning.

This paper studies two new mixed robust/average-case partitioning problems of the following form:

Prob. 1: $\max_{\pi \in \Pi} \Big[ \bar{\lambda} \min_i f_i(A_i^\pi) + \frac{\lambda}{m} \sum_{j=1}^m f_j(A_j^\pi) \Big]$, Prob. 2: $\min_{\pi \in \Pi} \Big[ \bar{\lambda} \max_i f_i(A_i^\pi) + \frac{\lambda}{m} \sum_{j=1}^m f_j(A_j^\pi) \Big]$,

where $0 \leq \lambda \leq 1$, $\bar{\lambda} \triangleq 1 - \lambda$, the set of sets $\pi = (A_1^\pi, A_2^\pi, \cdots, A_m^\pi)$ is a partition of a finite set $V$ (i.e, $\cup_i A_i^\pi = V$ and $\forall i \neq j, A_i^\pi \cap A_j^\pi = \emptyset$), and $\Pi$ refers to the set of all partitions of $V$ into $m$ blocks. The parameter $\lambda$ controls the objective: $\lambda = 1$ is the average case, $\lambda = 0$ is the robust case, and $0 < \lambda < 1$ is a mixed case. In general, Problems 1 and 2 are hopelessly intractable, even to approximate, but we assume that the $f_1, f_2, \cdots, f_m$ are all monotone non-decreasing (i.e., $f_i(S) \leq f_i(T)$ whenever $S \subseteq T$), normalized ($f_i(\emptyset) = 0$), and submodular [9] (i.e., $\forall S, T \subseteq V, f_i(S) + f_i(T) \geq f_i(S \cup T) + f_i(S \cap T)$). These assumptions allow us to develop fast, simple, and scalable algorithms that have approximation guarantees, as is done in this paper. These assumptions, moreover, allow us to retain the naturalness and applicability of Problems 1 and 2 to a wide variety of practical problems. Submodularity is a natural property in many real-world ML applications [20, 15, 18, 27]. When minimizing, submodularity naturally model notions of interacting costs and complexity, while when maximizing it readily models notions of diversity, summarization quality, and information. Hence, Problem 1 asks for a partition whose blocks each (and that collectively) are a good, say, summary of the whole. Problem 2 on the other hand, asks for a partition whose blocks each (and that collectively) are internally homogeneous (as is typical in clustering). Taken together, we call Problems 1 and 2 *Submodular Partitioning*. We further categorize these problems depending on if the $f_i$'s are identical to each other (*homogeneous*) or not (*heterogeneous*).[1] The heterogeneous case clearly generalizes the homogeneous setting, but as we will see, the additional homogeneous structure can be exploited to provide more efficient and/or tighter algorithms.

| Problem 1 (Max-(Min+Avg)) | | Problem 2 (Min-(Max+Avg)) | |
|---|---|---|---|
| | Approximation factor | | Approximation factor |
| $\lambda = 0$, BINSRCH [16] | $1/(2m - 1)$ | $\lambda = 0$, BALANCED$^\dagger$ [25] | $\min\{m, n/m\}$ |
| $\lambda = 0$, MATCHING [12] | $1/(n - m + 1)$ | $\lambda = 0$, SAMPLING [25] | $O(\sqrt{n} \log n)$ |
| $\lambda = 0$, ELLIPSOID [11] | $O(\sqrt{n} m^{\frac{1}{4}} \log n \log^{\frac{3}{2}} m)$ | $\lambda = 0$, ELLIPSOID [11] | $O(\sqrt{n} \log n)$ |
| $\lambda = 1$, GREEDWELFARE [8] | $1/2$ | $\lambda = 1$, GREEDSPLIT$^\dagger$ [29, 22] | $2$ |
| $\lambda = 0$, GREEDSAT* | $(1/2 - \delta, \frac{\delta}{1/2+\delta})$ | $\lambda = 1$, RELAX [5] | $O(\log n)$ |
| $\lambda = 0$, MMAX* | $O(\min_i \frac{1}{\|A_i^{\hat{\pi}}\| \sqrt{m} \log^3 m})$ | $\lambda = 0$, MMIN* | $2 \max_i \|A_i^{\pi^*}\|$ |
| $\lambda = 0$, GREEDMAX$^{\dagger *}$ | $1/m$ | $\lambda = 0$, LOVÁSZROUND* | $m$ |
| $0 < \lambda < 1$, COMBSFASWP* | $\max\{\frac{\beta\alpha}{\bar{\lambda}\beta+\alpha}, \lambda\beta\}$ | $0 < \lambda < 1$, COMBSLBSMP* | $\min\{\frac{m\alpha}{m\bar{\lambda}+\lambda}, \beta(m\bar{\lambda}+\lambda)\}$ |
| $0 < \lambda < 1$, GENERALGREEDSAT* | $\lambda/2$ | $0 < \lambda < 1$, GENERALLOVÁSZROUND* | $m$ |
| $\lambda = 0$, Hardness [12] | $1/2$ | $\lambda = 0$, Hardness* | $m$ |
| $\lambda = 1$, Hardness | $1 - 1/e$ [26] | $\lambda = 1$, Hardness | $2 - 2/m$ [7] |

Table 1: Summary of our contributions and existing work on Problems 1 and 2.[2] See text for details.

**Previous work:** Special cases of Problems 1 and 2 have appeared previously. Problem 1 with $\lambda = 0$ is called *submodular fair allocation* (SFA), and Problem 2 with $\lambda = 0$ is called *submodular load balancing* (SLB), robust optimization problems both of which previously have been studied. When $f_i$'s are all modular, SLB is called *minimum makespan scheduling*. An LP relaxation algorithm provides a 2-approximation for the heterogeneous setting [19]. When the objectives are submodular, the problem becomes much harder. Even in the homogeneous setting, [25] show that the problem is information theoretically hard to approximate within $o(\sqrt{n/\log n})$. They provide a balanced partitioning algorithm yielding a factor of $\min\{m, n/m\}$ under the homogeneous setting. They also give a sampling-based algorithm achieving $O(\sqrt{n/\log n})$ for the homogeneous setting. However, the sampling-based algorithm is not practical and scalable since it involves solving, in the worst-case, $O(n^3 \log n)$ instances of submodular function minimization each of which requires $O(n^5 \gamma + n^6)$ computation [23], where $\gamma$ is the cost of a function valuation. Another approach approximates each submodular function by its ellipsoid approximation (again non-scalable) and reduces SLB to its modular version (minimum makespan scheduling) leading to an approximation factor of $O(\sqrt{n} \log n)$ [11]. SFA, on the other hand, has been studied mostly in the heterogeneous setting. When $f_i$'s are all modular, the tightest algorithm, so far, is to iteratively round an LP solution achieving $O(1/(\sqrt{m} \log^3 m))$ approximation [2], whereas the problem is NP-hard to $1/2 + \epsilon$ approximate for any $\epsilon > 0$ [12]. When $f_i$'s are submodular, [12] gives a matching-based algorithm with a factor $1/(n - m + 1)$ approximation that performs poorly when $m \ll n$. [16] proposes a binary search algorithm yielding an improved factor of $1/(2m - 1)$. Similar to SLB, [11] applies the same ellipsoid

approximation techniques leading to a factor of $O(\sqrt{n}m^{1/4}\log n \log^{3/2} m)$. These approaches are theoretically interesting, but they do not scale to large problems. Problems 1 and 2, when $\lambda = 1$, have also been previously studied. Problem 2 becomes the *submodular multiway partition* (SMP) for which one can obtain a relaxation based 2-approximation [5] in the homogeneous case. In the heterogeneous case, the guarantee is $O(\log n)$ [6]. Similarly, [29, 22] propose a greedy splitting 2-approximation algorithm for the homogeneous setting. Problem 1 becomes the *submodular welfare* [26] for which a scalable greedy algorithm achieves a $1/2$ approximation [8]. Unlike the worst case ($\lambda = 0$), many of the algorithms proposed for these problems are scalable. The general case ($0 < \lambda < 1$) of Problems 1 and 2 differs from either of these extreme cases since we wish both for a *robust* (worst-case) and average case partitioning, and controlling $\lambda$ allows one to trade off between the two. As we shall see, the flexibility of a mixture can be more natural in certain applications.

**Applications:** There are a number of applications of submodular partitioning in ML as outlined below. Some of these we evaluate in Section 4. Submodular functions naturally capture notions of interacting cooperative costs and homogeneity and thus are useful for **clustering and image segmentation** [22, 17]. While the average case instance has been used before, a more worst-case variant (i.e., Problem 2 with $\lambda \approx 0$) is useful to produce **balanced clusterings** (i.e., the submodular valuations of all the blocks should be similar to each other). Problem 2 also addresses a problem in image segmentation, namely how to use only submodular functions (which are instances of pseudo-Boolean functions) for multi-label (i.e., non-Boolean) image segmentation. Problem 2 addresses this problem by allowing each segment $j$ to have its own submodular function $f_j$, and the objective measures the homogeneity $f_j(A_j^\pi)$ of segment $j$ based on the image pixels $A_j^\pi$ assigned to it. Moreover, by combining the average case and the worst case objectives, one can achieve a tradeoff between the two. Empirically, we evaluate our algorithms on unsupervised image segmentation (Section 4) and find that it outperforms other clustering methods including $k$-means, $k$-medoids, spectral clustering, and graph cuts.

Submodularity also accurately represents computational costs in **distributed systems**, as shown in [20]. In fact, [20] considers two separate problems: 1) text data partitioning for balancing memory demands; and 2) parameter partitioning for balancing communication costs. Both are treated by solving an instance of SLB (Problem 2, $\lambda = 0$) where memory costs are modeled using a set-cover submodular function and the communication costs are modeled using a modular (additive) function.

Another important ML application, evaluated in Section 4, is **distributed training** of statistical models. As data set sizes grow, the need for statistical training procedures tolerant of the **distributed data partitioning** becomes more important. Existing schemes are often developed and performed assuming data samples are distributed in an arbitrary or random fashion. As an alternate strategy, if the data is intelligently partitioned such that each block of samples can itself lead to a good approximate solution, a consensus amongst the distributed results could be reached more quickly than when under a poor partitioning. Submodular functions can in fact express the value of a subset of training data for certain machine learning risk functions, e.g., [27]. Using these functions within Problem 1, one can expect a partitioning (by formulating the problem as an instance of Problem 1, $\lambda \approx 0$) where each block is a good *representative* of the entire set, thereby achieving faster convergence in distributed settings. We demonstrate empirically, in Section 4, that this provides better results on several machine learning tasks, including the training of deep neural networks.

**Our Contributions:** In contrast to Problems 1 and 2 in the average case (i.e., $\lambda = 1$), existing algorithms for the worst case ($\lambda = 0$) are not scalable. This paper closes this gap, by proposing three new classes of algorithmic frameworks to solve SFA and SLB: (1) greedy algorithms; (2) semigradient-based algorithms; and (3) a Lovász extension based relaxation algorithm. For SFA, when $m = 2$, we formulate the problem as non-monotone submodular maximization, which can be approximated up to a factor of $1/2$ with $O(n)$ function evaluations [4]. For general $m$, we give a simple and scalable greedy algorithm (GREEDMAX), and show a factor of $1/m$ in the homogeneous setting, improving the state-of-the-art factor of $1/(2m-1)$ under the heterogeneous setting [16]. For the heterogeneous setting, we propose a "saturate" greedy algorithm (GREEDSAT) that iteratively solves instances of submodular welfare problems. We show GREEDSAT has a bi-criterion guarantee of $(1/2 - \delta, \delta/(1/2 + \delta))$, which ensures at least $\lceil m(1/2 - \delta) \rceil$ blocks receive utility at least $\delta/(1/2 + \delta)OPT$ for any $0 < \delta < 1/2$. For SLB, we first generalize the hardness result in [25] and show that it is hard to approximate better than $m$ for any $m = o(\sqrt{n/\log n})$ even in the homogeneous setting. We then give a Lovász extension based relaxation algorithm (LOVÁSZROUND) yielding a tight factor of $m$ for the heterogeneous setting. As far as we know, this is the first algorithm achieving a factor of $m$ for SLB in this setting. For both SFA and SLB, we also obtain more efficient

algorithms with bounded approximation factors, which we call majorization-minimization (MMIN) and minorization-maximization (MMAX).

Next we show algorithms to handle Problems 1 and 2 with general $0 < \lambda < 1$. We first give two simple and generic schemes (COMBSFASWP and COMBSLBSMP), both of which efficiently combines an algorithm for the worst-case problem (special case with $\lambda = 0$), and an algorithm for the average case (special case with $\lambda = 1$) to provide a guarantee interpolating between the two bounds. For Problem 1 we generalize GREEDSAT leading to GENERALGREEDSAT, whose guarantee smoothly interpolates in terms of $\lambda$ between the bi-criterion factor by GREEDSAT in the case of $\lambda = 0$ and the constant factor of $1/2$ by the greedy algorithm in the case of $\lambda = 1$. For Problem 2 we generalize LOVÁSZROUND to obtain a relaxation algorithm (GENERALLOVÁSZROUND) that achieves an $m$-approximation for general $\lambda$. The theoretical contributions and the existing work for Problems 1 and 2 are summarized in Table 1.

Lastly, we demonstrate the efficacy of Problem 2 on unsupervised image segmentation, and the success of Problem 1 to distributed machine learning, including ADMM and neural network training.

## 2 Robust Submodular Partitioning (Problems 1 and 2 when $\lambda = 0$)

Notation: we define $f(j|S) \triangleq f(S \cup j) - f(S)$ as the gain of $j \in V$ in the context of $S \subseteq V$. We assume w.l.o.g. that the ground set is $V = \{1, 2, \cdots, n\}$.

### 2.1 Approximation Algorithms for SFA (Problem 1 with $\lambda = 0$)

We first study approximation algorithms for SFA. When $m = 2$, the problem becomes $\max_{A \subseteq V} g(A)$ where $g(A) = \min\{f_1(A), f_2(V \setminus A)\}$ and is submodular thanks to Theorem 2.1.

**Theorem 2.1.** *If $f_1$ and $f_2$ are monotone submodular, $\min\{f_1(A), f_2(V \setminus A)\}$ is also submodular.*

Proofs for all theorems in this paper are given in [28]. The simple bi-directional randomized greedy algorithm [4] therefore approximates SFA with $m = 2$ to a factor of $1/2$ matching the problem's hardness. For general $m$, we approach SFA from the perspective of the greedy algorithms. In this work we introduce two variants of a greedy algorithm – GREEDMAX (Alg. 1) and GREEDSAT (Alg. 2), suited to the homogeneous and heterogeneous settings, respectively.

**GREEDMAX:** The key idea of GREEDMAX (see Alg. 1) is to greedily add an item with the maximum marginal gain to the block whose current solution is minimum. Initializing $\{A_i\}_{i=1}^m$ with the empty sets, the greedy flavor also comes from that it incrementally grows the solution by greedily improving the overall objective $\min_{i=1,\dots,m} f_i(A_i)$ until $\{A_i\}_{i=1}^m$ forms a partition. Besides its simplicity, Theorem 2.2 offers the optimality guarantee.

**Theorem 2.2.** GREEDMAX *achieves a guarantee of $1/m$ under the homogeneous setting.*

By assuming the homogeneity of the $f_i$'s, we obtain a very simple $1/m$-approximation algorithm improving upon the state-of-the-art factor $1/(2m-1)$ [16]. Thanks to the lazy evaluation trick as described in [21], Line 5 in Alg. 1 need not to recompute the marginal gain for every item in each round, leading GREEDMAX to scale to large data sets.

**GREEDSAT:** Though simple and effective in the homogeneous setting, GREEDMAX performs arbitrarily poorly under the heterogeneous setting. To this end we provide another algorithm – "Saturate" Greedy (GREEDSAT, see Alg. 2). The key idea of GREEDSAT is to relax SFA to a much simpler problem – Submodular Welfare (SWP), i.e., Problem 1 with $\lambda = 0$. Similar in flavor to the one proposed in [18] GREEDSAT defines an intermediate objective $\bar{F}^c(\pi) = \sum_{i=1}^m f_i^c(A_i^\pi)$, where $f_i^c(A) = \frac{1}{m} \min\{f_i(A), c\}$ (Line 2). The parameter $c$ controls the saturation in each block. $f_i^c$ satisfies submodularity for each $i$. Unlike SFA, the combinatorial optimization problem $\max_{\pi \in \Pi} \bar{F}^c(\pi)$ (Line 6) is much easier and is an instance of SWP. In this work, we solve Line 6 by the efficient greedy algorithm as described in [8] with a factor $1/2$. One can also use a more computationally expensive multi-linear relaxation algorithm as given in [26] to solve Line 6 with a tight factor $\alpha = (1 - 1/e)$. Setting the input argument $\alpha$ as the approximation factor for Line 6, the essential idea of GREEDSAT is to perform a binary search over the parameter $c$ to find the largest $c^*$ such that the returned solution $\hat{\pi}^{c^*}$ for the instance of SWP satisfies $\bar{F}^{c^*}(\hat{\pi}^{c^*}) \geq \alpha c^*$. GREEDSAT terminates after solving $O(\log(\frac{\min_i f_i(V)}{\epsilon}))$ instances of SWP. Theorem 2.3 gives a bi-criterion optimality guarantee.

**Theorem 2.3.** *Given $\epsilon > 0$, $0 \leq \alpha \leq 1$ and any $0 < \delta < \alpha$, GREEDSAT finds a partition such that at least $\lceil m(\alpha - \delta) \rceil$ blocks receive utility at least $\frac{\delta}{1-\alpha+\delta}(\max_{\pi \in \Pi} \min_i f_i(A_i^\pi) - \epsilon)$.*

**Algorithm 1: GREEDMAX**

1: Input: $f, m, V$.
2: Let $A_1 =, \ldots, = A_m = \emptyset$; $R = V$.
3: **while** $R \neq \emptyset$ **do**
4: $\quad j^* \in \mathrm{argmin}_j f(A_j)$;
5: $\quad a^* \in \mathrm{argmax}_{a \in R} f(a|A_{j^*})$
6: $\quad A_{j^*} \leftarrow A_{j^*} \cup \{a^*\}$; $R \leftarrow R \setminus a^*$
7: **end while**
8: Output $\{A_i\}_{i=1}^m$.

**Algorithm 2: GREEDSAT**

1: Input: $\{f_i\}_{i=1}^m, m, V, \alpha$.
2: Let $\bar{F}^c(\pi) = \frac{1}{m} \sum_{i=1}^m \min\{f_i(A_i^\pi), c\}$.
3: Let $c_{\min} = 0, c_{\max} = \min_i f_i(V)$
4: **while** $c_{\max} - c_{\min} \geq \epsilon$ **do**
5: $\quad c = \frac{1}{2}(c_{\max} + c_{\min})$
6: $\quad \hat{\pi}^c \in \mathrm{argmax}_{\pi \in \Pi} \bar{F}^c(\pi)$
7: $\quad$ **if** $\bar{F}^c(\hat{\pi}^c) < \alpha c$ **then**
8: $\quad\quad c_{\max} = c$
9: $\quad$ **else**
10: $\quad\quad c_{\min} = c$; $\hat{\pi} \leftarrow \hat{\pi}^c$
11: $\quad$ **end if**
12: **end while**
13: Output: $\hat{\pi}$.

**Algorithm 3: LOVÁSZROUND**

1: Input: $\{f_i\}_{i=1}^m, \{\tilde{f}_i\}_{i=1}^m, m, V$.
2: Solve for $\{x_i^*\}_{i=1}^m$ via convex relaxation.
3: Rounding: Let $A_1 =, \ldots, = A_m = \emptyset$.
4: **for** $j = 1, \ldots, n$ **do**
5: $\quad \hat{i} \in \mathrm{argmax}_i x_i^*(j)$; $A_{\hat{i}} = A_{\hat{i}} \cup j$
6: **end for**
7: Output $\{A_i\}_{i=1}^m$.

**Algorithm 4: GREEDMIN**

1: Input: $f, m, V$;
2: Let $A_1 =, \ldots, = A_m = \emptyset$; $R = V$.
3: **while** $R \neq \emptyset$ **do**
4: $\quad j^* \in \mathrm{argmin}_j f(A_j)$
5: $\quad a^* \in \min_{a \in R} f(a|A_{j^*})$
6: $\quad A_{j^*} \leftarrow A_{j^*} \cup a^*$; $R \leftarrow R \setminus a^*$
7: **end while**
8: Output $\{A_i\}_{i=1}^m$.

**Algorithm 5: MMIN**

1: Input: $\{f_i\}_{i=1}^m, m, V$, partition $\pi^0$.
2: Let $t = 0$
3: **repeat**
4: $\quad$ **for** $i = 1, \ldots, m$ **do**
5: $\quad\quad$ Pick a supergradient $m_i$ at $A_i^{\pi^t}$ for $f_i$.
6: $\quad$ **end for**
7: $\quad \pi^{t+1} \in \mathrm{argmin}_{\pi \in \Pi} \max_i m_i(A_i^\pi)$
8: $\quad t = t + 1$;
9: **until** $\pi^t = \pi^{t-1}$
10: Output: $\pi^t$.

**Algorithm 6: MMAX**

1: Input: $\{f_i\}_{i=1}^m, m, V$, partition $\pi^0$.
2: Let $t = 0$.
3: **repeat**
4: $\quad$ **for** $i = 1, \ldots, m$ **do**
5: $\quad\quad$ Pick a subgradient $h_i$ at $A_i^{\pi^t}$ for $f_i$.
6: $\quad$ **end for**
7: $\quad \pi^{t+1} \in \mathrm{argmax}_{\pi \in \Pi} \min_i h_i(A_i^\pi)$
8: $\quad t = t + 1$;
9: **until** $\pi^t = \pi^{t-1}$
10: Output: $\pi^t$.

For any $0 < \delta < \alpha$ Theorem 2.3 ensures that the top $\lceil m(\alpha - \delta) \rceil$ valued blocks in the partition returned by GREEDSAT are $(\delta/(1-\alpha+\delta)-\epsilon)$-optimal. $\delta$ controls the trade-off between the number of top valued blocks to bound and the performance guarantee attained for these blocks. The smaller $\delta$ is, the more top blocks are bounded, but with a weaker guarantee. We set the input argument $\alpha = 1/2$ (or $\alpha = 1 - 1/e$) as the worst-case performance guarantee for solving SWP so that the above theoretical analysis follows. However, the worst-case is often achieved only by very contrived submodular functions. For the ones used in practice, the greedy algorithm often leads to near-optimal solution ([18] and our own observations). Setting $\alpha$ as the actual performance guarantee for SWP (often very close to 1) can improve the empirical bound, and we, in practice, typically set $\alpha = 1$ to good effect.

**MMAX**: Lastly, we introduce another algorithm for the heterogeneous setting, called minorization-maximization (MMAX, see Alg. 6). Similar to the one proposed in [14], the idea is to iteratively maximize tight lower bounds of the submodular functions. Submodular functions have tight modular lower bounds, which are related to the subdifferential $\partial_f(Y)$ of the submodular set function $f$ at a set $Y \subseteq V$ [9]. Denote a subgradient at $Y$ by $h_Y \in \partial_f(Y)$, the extreme points of $\partial_f(Y)$ may be computed via a greedy algorithm: Let $\sigma$ be a permutation of $V$ that assigns the elements in $Y$ to the first $|Y|$ positions ($\sigma(i) \in Y$ if and only if $i \leq |Y|$). Each such permutation defines a chain with elements $S_0^\sigma = \emptyset, S_i^\sigma = \{\sigma(1), \sigma(2), \ldots, \sigma(i)\}$, and $S_{|Y|}^\sigma = Y$. An extreme point $h_Y^\sigma$ of $\partial_f(Y)$ has each entry as $h_Y^\sigma(\sigma(i)) = f(S_i^\sigma) - f(S_{i-1}^\sigma)$. Defined as above, $h_Y^\sigma$ forms a lower bound of $f$, tight at $Y$ — i.e., $h_Y^\sigma(X) = \sum_{j \in X} h_Y^\sigma(j) \leq f(X), \forall X \subseteq V$ and $h_Y^\sigma(Y) = f(Y)$. The idea of MMAX is to consider a modular lower bound tight at the set corresponding to each block of a partition. In other words, at iteration $t + 1$, for each block $i$, we approximate $f_i$ with its modular lower bound tight at $A_i^{\pi^t}$ and solve a modular version of Problem 1 (Line 7), which admits efficient approximation algorithms [2]. MMAX is initialized with a partition $\pi^0$, which is obtained by solving Problem 1, where each $f_i$ is replaced with a simple modular function $f_i'(A) = \sum_{a \in A} f_i(a)$. The following worst-case bound holds:

**Theorem 2.4.** MMAX *achieves a worst-case guarantee of* $O\left(\min_i \frac{1+(|A_i^{\hat{\pi}}|-1)(1-\kappa_{f_i}(A_i^{\hat{\pi}}))}{|A_i^{\hat{\pi}}|\sqrt{m}\log^3 m}\right)$, *where*

$\hat{\pi} = (A_1^{\hat{\pi}}, \cdots, A_m^{\hat{\pi}})$ *is the partition obtained by the algorithm, and* $\kappa_f(A) = 1 - \min_{v \in V} \frac{f(v|A \setminus v)}{f(v)} \in [0, 1]$ *is the curvature of a submodular function* $f$ *at* $A \subseteq V$.

## 2.2 Approximation Algorithms for SLB (Problem 2 with $\lambda = 0$)

We next investigate SLB, where existing hardness results [25] are $o(\sqrt{n/\log n})$, which is independent of $m$ and implicitly assumes that $m = \Theta(\sqrt{n/\log n})$. However, applications for SLB are often dependent on $m$ with $m \ll n$. We hence offer hardness analysis in terms of $m$ in the following.

**Theorem 2.5.** *For any $\epsilon > 0$, SLB cannot be approximated to a factor of $(1 - \epsilon)m$ for any $m = o(\sqrt{n/\log n})$ with polynomial number of queries even under the homogeneous setting.*

For the rest of the paper, we assume $m = o(\sqrt{n/\log n})$ for SLB, unless stated otherwise.

**GREEDMIN:** Theorem 2.5 implies that SLB is hard to approximate better than $m$. However, an arbitrary partition $\pi \in \Pi$ already achieves the best approximation factor of $m$ that one can hope for under the homogeneous setting, since $\max_i f(A_i^\pi) \leq f(V) \leq \sum_i f(A_i^{\pi'}) \leq m \max_i f(A_i^{\pi'})$ for any $\pi' \in \Pi$. In practice, one can still implement a greedy style heuristic, which we refer to as GREEDMIN (Alg. 4). Very similar to GREEDMAX, GREEDMIN only differs in Line 5, where the item with the smallest marginal gain is added. Since the functions are all monotone, any additions to a block can (if anything) only increase its value, so we choose to add to the minimum valuation block in Line 4 to attempt to keep the maximum valuation block from growing further.

**LOVÁSZ ROUND:** Next we consider the heterogeneous setting, for which we propose a tight algorithm – LOVÁSZROUND (see Alg. 3). The algorithm proceeds as follows: (1) apply the Lovász extension of submodular functions to relax SLB to a convex program, which is exactly solved to a fractional solution (Line 2); (2) map the fractional solution to a partition using the $\theta$-rounding technique as proposed in [13] (Line 3 - 6). The Lovász extension, which naturally connects a submodular function $f$ with its convex relaxation $\tilde{f}$, is defined as follows: given any $x \in [0,1]^n$, we obtain a permutation $\sigma_x$ by ordering its elements in non-increasing order, and thereby a chain of sets $S_0^{\sigma_x} \subset, \ldots, \subset S_n^{\sigma_x}$ with $S_j^{\sigma_x} = \{\sigma_x(1), \ldots, \sigma_x(j)\}$ for $j = 1, \ldots, n$. The Lovász extension $\tilde{f}$ for $f$ is the weighted sum of the ordered entries of $x$: $\tilde{f}(x) = \sum_{j=1}^n x(\sigma_x(j))(f(S_j^{\sigma_x}) - f(S_{j-1}^{\sigma_x}))$. Given the convexity of the $\tilde{f}_i$'s , SLB is relaxed to the following convex program:

$$\min_{x_1,\ldots,x_m \in [0,1]^n} \max_i \tilde{f}_i(x_i), \text{ s.t } \sum_{i=1}^m x_i(j) \geq 1, \text{ for } j = 1, \ldots, n \tag{1}$$

Denoting the optimal solution for Eqn 1 as $\{x_1^*, \ldots, x_m^*\}$, the $\theta$-rounding step simply maps each item $j \in V$ to a block $\hat{i}$ such that $\hat{i} \in \arg\max_i x_i^*(j)$ . The bound for LOVÁSZROUND is as follows:

**Theorem 2.6.** LOVÁSZROUND *achieves a worst-case approximation factor $m$.*

We remark that, to the best of our knowledge, LOVÁSZROUND is the first algorithm that is tight and that gives an approximation in terms of $m$ for the heterogeneous setting.

**MMIN:** Similar to MMAX for SFA, we propose Majorization-Minimization (MMIN, see Alg. 5) for SLB. Here, we iteratively choose modular upper bounds, which are defined via superdifferentials $\partial^f(Y)$ of a submodular function [15] at $Y$. Moreover, there are specific supergradients [14] that define the following two modular upper bounds (when referring to either one, we use $m_X^f$):

$$m_{X,1}^f(Y) \triangleq f(X) - \sum_{j \in X \setminus Y} f(j|X \setminus j) + \sum_{j \in Y \setminus X} f(j|\emptyset), \quad m_{X,2}^f(Y) \triangleq f(X) - \sum_{j \in X \setminus Y} f(j|V \setminus j) + \sum_{j \in Y \setminus X} f(j|X).$$

Then $m_{X,1}^f(Y) \geq f(Y)$ and $m_{X,2}^f(Y) \geq f(Y), \forall Y \subseteq V$ and $m_{X,1}^f(X) = m_{X,2}^f(X) = f(X)$. At iteration $t + 1$, for each block $i$, MMIN replaces $f_i$ with a choice of its modular upper bound $m_i$ tight at $A_i^{\pi^t}$ and solves a modular version of Problem 2 (Line 7), for which there exists an efficient LP relaxation based algorithm [19]. Similar to MMAX, the initial partition $\pi^0$ is obtained by solving Problem 2, where each $f_i$ is substituted with $f_i'(A) = \sum_{a \in A} f_i(a)$. The following worst-case bound holds:

**Theorem 2.7.** MMIN *achieves a worst-case guarantee of* $(2 \max_i \frac{|A_i^{\pi^*}|}{1 + (|A_i^{\pi^*}| - 1)(1 - \kappa_{f_i}(A_i^{\pi^*}))})$, *where* $\pi^* = (A_1^{\pi^*}, \cdots, A_m^{\pi^*})$ *denotes the optimal partition.*

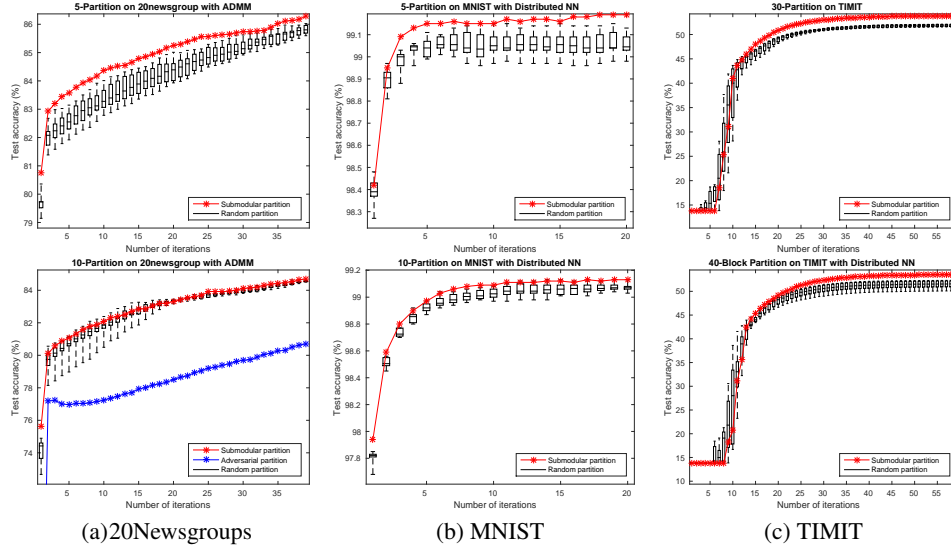

Figure 1: Comparison between submodular and random partitions for distributed ML, including ADMM (Fig 1a) and distributed neural nets (Fig 1b) and (Fig 1c). For the box plots, the central mark is the median, the box edges are 25th and 75th percentiles, and the bars denote the best and worst cases.

## 3 General Submodular Partitioning (Problems 1 and 2 when $0 < \lambda < 1$)

In this section we study Problem 1 and Problem 2, in the most general case, i.e., $0 < \lambda < 1$. We first propose a simple and general "extremal combination" scheme that works both for problem 1 and 2. It naturally combines an algorithm for solving the worst-case problem ($\lambda = 0$) with an algorithm for solving the average case ($\lambda = 1$). We use Problem 1 as an example, but the same scheme easily works for Problem 2. Denote ALGWC as the algorithm for the worst-case problem (i.e. SFA), and ALGAC as the algorithm for the average case (i.e., SWP). The scheme is to first obtain a partition $\hat{\pi}_1$ by running ALGWC on the instance of Problem 1 with $\lambda = 0$ and a second partition $\hat{\pi}_2$ by running ALGAC with $\lambda = 1$. Then we output one of $\hat{\pi}_1$ and $\hat{\pi}_2$, with which the higher valuation for Problem 1 is achieved. We call this scheme COMBSFASWP. Suppose ALGWC solves the worst-case problem with a factor $\alpha \leq 1$ and ALGAC for the average case with $\beta \leq 1$. When applied to Problem 2 we refer to this scheme as COMBSLBSMP ($\alpha \geq 1$ and $\beta \geq 1$). The following guarantee holds for both schemes:

**Theorem 3.1.** *For any $\lambda \in (0,1)$ COMBSFASWP solves Problem 1 with a factor* $\max\{\frac{\beta\alpha}{\lambda\beta+\alpha}, \lambda\beta\}$ *in the* heterogeneous *case, and* $\max\{\min\{\alpha, \frac{1}{m}\}, \frac{\beta\alpha}{\lambda\beta+\alpha}, \lambda\beta\}$ *in the* homogeneous *case. Similarly,* COMBSLBSMP *solves Problem 2 with a factor* $\min\{\frac{m\alpha}{m\bar{\lambda}+\lambda}, \beta(m\bar{\lambda}+\lambda)\}$ *in the* heterogeneous *case, and* $\min\{m, \frac{m\alpha}{m\bar{\lambda}+\lambda}, \beta(m\bar{\lambda}+\lambda)\}$ *in the* homogeneous *case.*

The drawback of COMBSFASWP and COMBSLBSMP is that they do not explicitly exploit the trade-off between the average-case and worst-case objectives in terms of $\lambda$. To obtain more practically interesting algorithms, we also give GENERALGREEDSAT that generalizes GREEDSAT to solve Problem 1. Similar to GREEDSAT we define an intermediate objective: $\bar{F}_\lambda^c(\pi) = \frac{1}{m}\sum_{i=1}^m \min\{\bar{\lambda}f_i(A_i^\pi) + \lambda\frac{1}{m}\sum_{j=1}^m f_j(A_j^\pi), c\}$ in GENERALGREEDSAT. Following the same algorithmic design as in GREEDSAT, GENERALGREEDSAT only differs from GREEDSAT in Line 6, where the submodular welfare problem is defined on the new objective $\bar{F}_\lambda^c(\pi)$. In [28] we show that GENERALGREEDSAT gives $\lambda/2$ approximation, while also yielding a bi-criterion guarantee that generalizes Theorem 2.3. In particular GENERALGREEDSAT recovers the bicriterion guarantee as shown in Theorem 2.3 when $\lambda = 0$. In the case of $\lambda = 1$, GENERALGREEDSAT recovers the $1/2$-approximation guarantee of the greedy algorithm for solving the submodular welfare problem, i.e., the average-case objective. Moreover an improved guarantee is achieved by GENERALGREEDSAT as $\lambda$ increases. Details are given in [28].

To solve Problem 2 we generalize LOVÁSZROUND leading to GENERALLOVÁSZROUND. Similar to LOVÁSZROUND we relax each submodular objective as its convex relaxation using the Lovász extension. Almost the same as LOVÁSZROUND, GENERALLOVÁSZROUND only differs in Line 2, where Problem 2 is relaxed as the following convex program: $\min_{x_1,\dots,x_m \in [0,1]^n} \bar{\lambda}\max_i \tilde{f}_i(x_i) +$

$\lambda\frac{1}{m}\sum_{j=1}^{m}\tilde{f}_j(x_j)$, s.t $\sum_{i=1}^{m}x_i(j)\geq 1$, for $j=1,\ldots,n$. Following the same rounding procedure as LOVÁSZ ROUND, GENERALLOVÁSZ ROUND is guaranteed to give an $m$-approximation for Problem 2 with general $\lambda$. Details are given in [28].

## 4 Experiments and Conclusions

We conclude in this section by empirically evaluating the algorithms proposed for Problems 1 and 2 on real-world data partitioning applications including distributed ADMM, distributed deep neural network training, and lastly unsupervised image segmentation tasks.

**ADMM:** We first consider data partitioning for distributed convex optimization. The evaluation task is text categorization on the 20 Newsgroup data set, which consists of 18,774 articles divided almost evenly across 20 classes. We formulate the multi-class classification as an $\ell_2$ regularized logistic regression, which is solved by ADMM implemented as [3]. We run 10 instances of random partitioning on the training data as a baseline. In this case, we use the feature based function (same as the one used in [27]), in the homogeneous setting of Problem 1 (with $\lambda=0$). We use GREEDMAX as the partitioning algorithm. In Figure 1a, we observe that the resulting partitioning performs much better than a random partitioning (and significantly better than an adversarial partitioning, formed by grouping similar items together). More details are given in [28].

**Distributed Deep Neural Network (DNN) Training:** Next we evaluate our framework on distributed deep neural network (DNN) training. We test on two tasks: 1) handwritten digit recognition on the MNIST database, which consists of 60,000 training and 10,000 test samples; 2) phone classification on the TIMIT data, which has 1,124,823 training and 112,487 test samples. A 4-layer DNN model is applied to the MNIST experiment, and we train a 5-layer DNN for TIMIT. For both experiments the submodular partitioning is obtained by solving the homogeneous case of Problem 1 ($\lambda=0$) using GREEDMAX on a form of clustered facility location (as proposed and used in [27]). We perform distributed training using an averaging stochastic gradient descent scheme, similar to the one in [24]. We also run 10 instances of random partitioning as a baseline. As shown in Figure 1b and 1c, the submodular partitioning outperforms the random baseline. An adversarial partitioning, which is formed by grouping items with the same class, in either case, cannot even be trained.

**Unsupervised Image Segmentation:** We test the efficacy of Problem 2 on unsupervised image segmentation over the GrabCut data set (30 color images and their ground truth foreground/background labels). By "unsupervised", we mean that no labeled data at any time in supervised or semi-supervised training, nor any kind of interactive segmentation, was used in forming or optimizing the objective. The submodular partitioning for each image is obtained by solving the

| F-measure on all of GrabCut | | | | | | | | | |
|---|---|---|---|---|---|---|---|---|---|
| | Original | 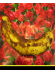 |  |  |  |  |  |  |  |
| 1.0 | Ground Truth |  |  |  |  |  |  |  |  |
| 0.810 | k-means |  |  |  |  |  |  |  |  |
| 0.823 | k-medoids |  |  |  |  |  |  |  |  |
| 0.854 | Spectral Clustering |  |  |  |  |  |  |  |  |
| 0.853 | Graph Cut |  |  |  |  |  |  |  |  |
| **0.870** | Submodular Partitioning |  |  |  |  |  |  |  |  |

Figure 2: Unsupervised image segmentation (right: some examples).

homogeneous case of Problem 2 ($\lambda=0.8$) using a modified variant of GREEDMIN on the facility location function. We compare our method against the other unsupervised methods $k$-means, $k$-medoids, spectral clustering, and graph cuts. Given an $m$-partition of an image and its ground truth labels, we assign each of the $m$ blocks either to the foreground or background label having the larger intersection. In Fig. 2 we show example segmentation results after this mapping on several example images as well as averaged F-measure (relative to ground truth) over the whole data set. More details are given in [28].

**Acknowledgments:** This material is based upon work supported by the National Science Foundation under Grant No. IIS-1162606, the National Institutes of Health under award R01GM103544, and by a Google, a Microsoft, and an Intel research award. R. Iyer acknowledges support from a Microsoft Research Ph.D Fellowship. This work was supported in part by TerraSwarm, one of six centers of STARnet, a Semiconductor Research Corporation program sponsored by MARCO and DARPA.

## Footnotes

[1]Similar sub-categorizations have been called the "uniform" vs. the "non-uniform" case in the past [25, 11].

[2]Results obtained in this paper are marked as *. Methods for only the homogeneous setting are marked as †.

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
