[Supplementary Material · submod_partitioning_extended.pdf]

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

## 1.1 Sub-categorizations and Related Previous Work

**Problem 1:** Special cases of Problem 1 have appeared previously in the literature. Problem 1 with $\lambda = 0$ is called *submodular fair allocation* (SFA), which has been studied mostly in the heterogeneous setting. When $f_i$'s are all modular, the tightest algorithm, so far, is to iteratively round an LP solution achieving $O(1/(\sqrt{m} \log^3 m))$ approximation [2], whereas the problem is NP-hard to $1/2 + \epsilon$ approximate for any $\epsilon > 0$ [18]. When $f_i$'s are submodular, [18] gives a matching-based algorithm with a factor $1/(n - m + 1)$ approximation that performs poorly when $m \ll n$. [27] propose a binary search algorithm yielding an improved factor of $1/(2m - 1)$. Another approach approximates each submodular function by its ellipsoid approximation (non-scalable) and reduces SFA to its modular version leading to an approximation factor of $O(\sqrt{n} m^{1/4} \log n \log^{3/2} m)$. These approaches are theoretically interesting, but they either do not fully exploit the problem structure or cannot scale to large problems. On the other hand, Problem 1 for $\lambda = 1$ is called *submodular welfare*. This problem has been extensively studied in the literature and can be equivalently formulated as submodular maximization under a partition matroid constraint [44]. It admits a scalable greedy algorithm that achieves a $1/2$ approximation [14]. More recently a multi-linear extension based algorithm nicely solves the submodular welfare problem with a factor of $(1 - 1/e)$ matching the hardness of this problem [44]. As far as we know, Problem 1 for general $0 < \lambda < 1$ has not been studied in the literature.

**Problem 2:** When $\lambda = 0$, Problem 2 is studied as *submodular load balancing* (SLB). When $f_i$'s are all modular, SLB is called *minimum makespan scheduling*. In the homogeneous setting, [19] give a PTAS scheme ($(1 + \epsilon)$-approximation algorithm which runs in polynomial time for any fixed $\epsilon$), while an LP relaxation algorithm provides a 2-approximation for the heterogeneous setting [32]. When the objectives are submodular, the problem becomes much harder. Even in the homogeneous setting, [40]

| Problem 1 (Max-(Min+Avg)) | |
| --- | --- |
| | Approximation factor |
| $\lambda = 0$, BINSRCH [27] | $1/(2m-1)$ |
| $\lambda = 0$, MATCHING [18] | $1/(n-m+1)$ |
| $\lambda = 0$, ELLIPSOID [17] | $O(\sqrt{n}m^{1/4}\log n \log^{3/2} m)$ |
| $\lambda = 1$, GREEDWELFARE [14] | $1/2$ |
| $\lambda = 0$, GREEDSAT* | $(1/2-\delta, \frac{\delta}{1/2+\delta})$ |
| $\lambda = 0$, MMAX* | $O(\min_i \frac{1+(|A_i^{\hat{\pi}}|-1)(1-\kappa_{f_i}(A_i^{\hat{\pi}}))}{|A_i^{\hat{\pi}}|\sqrt{m}\log^3 m})$ |
| $\lambda = 0$, GREEDMAX$^{\dagger *}$ | $1/m$ |
| $0 < \lambda < 1$, GENERALGREEDSAT* | $\lambda/2$ |
| $0 < \lambda < 1$, COMBSFASWP* | $\max\{\frac{\beta\alpha}{\lambda\beta+\alpha}, \lambda\beta\}$ |
| $0 < \lambda < 1$, COMBSFASWP$^{\dagger *}$ | $\max\{\min\{\alpha, \frac{1}{m}\}, \frac{\beta\alpha}{\lambda\beta+\alpha}, \lambda\beta\}$ |
| $\lambda = 0$, Hardness | $1/2$ [18] |
| $\lambda = 1$, Hardness | $1-1/e$ [44] |

Table 1: Summary of our contributions and existing work on Problem 1.[2] See text for details.

show that the problem is information theoretically hard to approximate within $o(\sqrt{n/\log n})$. They provide a balanced partitioning algorithm yielding a factor of $\min\{m, n/m\}$ under the homogeneous setting. They also give a sampling-based algorithm achieving $O(\sqrt{n/\log n})$ for the homogeneous setting. However, the sampling-based algorithm is not practical and scalable since it involves solving, in the worst-case, $O(n^3 \log n)$ instances of submodular function minimization each of which in general currently requires $O(n^5\gamma + n^6)$ computation [37], where $\gamma$ is the cost of a function valuation. Similar to Submodular Fair Allocation, [17] applies the same ellipsoid approximation techniques leading to a factor of $O(\sqrt{n}\log n)$ [17]. When $\lambda = 1$, Problem 2 becomes the *submodular multiway partition* (SMP) for which one can obtain a relaxation based 2-approximation [8] in the homogeneous case. In the non-homogeneous case, the guarantee is $O(\log n)$ [9]. Similarly, [50, 36] propose a greedy splitting 2-approximation algorithm for the homogeneous setting. To the best of our knowledge, there does not exist any work on Problem 2 with general $0 < \lambda < 1$.

| Problem 2 (Min-(Max+Avg)) | |
| --- | --- |
| | Approximation factor |
| $\lambda = 0$, BALANCED$^{\dagger}$ [40] | $\min\{m, n/m\}$ |
| $\lambda = 0$, SAMPLING [40] | $O(\sqrt{n}\log n)$ |
| $\lambda = 0$, ELLIPSOID [17] | $O(\sqrt{n}\log n)$ |
| $\lambda = 1$, GREEDSPLIT$^{\dagger}$ [50, 36] | $2$ |
| $\lambda = 1$, RELAX [8] | $O(\log n)$ |
| $\lambda = 0$, MMIN* | $\max_i \frac{2|A_i^{\pi^*}|}{1+(|A_i^{\pi^*}|-1)(1-\kappa_{f_i}(A_i^{\pi^*}))}$ |
| $\lambda = 0$, LOVÁSZROUND* | $m$ |
| $0 < \lambda < 1$ GENERALLOVÁSZROUND* | $m$ |
| $0 < \lambda < 1$, COMBSLBSMP* | $\min\{\frac{m\alpha}{m\lambda+\lambda}, \beta(m\lambda+\lambda)\}$ |
| $0 < \lambda < 1$, COMBSLBSMP$^{\dagger *}$ | $\min\{m, \frac{m\alpha}{m\lambda+\lambda}, \beta(m\lambda+\lambda)\}$ |
| $\lambda = 0$, Hardness* | $m$ |
| $\lambda = 1$, Hardness | $2-2/m$ [12] |

Table 2: Summary of our contributions and existing work on Problem 2.[3]

## 1.2 Our Contributions

In contrast to Problems 1 and 2 in the average case (i.e., $\lambda = 1$), existing algorithms for the worst case ($\lambda = 0$) are not scalable. This paper closes this gap, by proposing three new classes of algorithmic

frameworks to solve SFA and SLB: (1) greedy algorithms; (2) semigradient-based algorithms; and (3) a Lovász extension based relaxation algorithm.

For SFA, when $m = 2$, we formulate the problem as non-monotone submodular maximization, which can be approximated up to a factor of $1/2$ with $O(n)$ function evaluations [7]. For general $m$, we give a simple and scalable greedy algorithm (GREEDMAX), and show a factor of $1/m$ in the homogeneous setting, improving the state-of-the-art factor of $1/(2m - 1)$ under the heterogeneous setting [27]. For the heterogeneous setting, we propose a "saturate" greedy algorithm (GREEDSAT) that iteratively solves instances of submodular welfare problems. We show GREEDSAT has a bi-criterion guarantee of $(1/2 - \delta, \delta/(1/2 + \delta))$, which ensures at least $\lceil m(1/2 - \delta) \rceil$ blocks receive utility at least $\delta/(1/2 + \delta)OPT$ for any $0 < \delta < 1/2$. For SLB, we first generalize the hardness result in [40] and show that it is hard to approximate better than $m$ for any $m = o(\sqrt{n/\log n})$ even in the homogeneous setting. We then give a Lovász extension based relaxation algorithm (LOVÁSZ ROUND) yielding a tight factor of $m$ for the heterogeneous setting. As far as we know, this is the first algorithm achieving a factor of $m$ for SLB in this setting. For both SFA and SLB, we also obtain more efficient algorithms with bounded approximation factors, which we call majorization-minimization (MMIN) and minorization-maximization (MMAX).

Next we show algorithms to handle Problems 1 and 2 with general $0 < \lambda < 1$. We first give two simple and generic schemes (COMBSFASWP and COMBSLBSMP), both of which efficiently combines an algorithm for the worst-case problem (special case with $\lambda = 0$), and an algorithm for the average case (special case with $\lambda = 1$) to provide a guarantee interpolating between the two bounds. Given the efficient algorithms proposed in this paper for the robust (worst case) problems (with guarantee $\alpha$), and an existing algorithm for the average case (say, with a guarantee $\beta$), we can obtain a combined guarantee in terms of $\alpha, \beta$ and $\lambda$. We then generalize the proposed algorithms for SLB and SFA to give more practical algorithmic frameworks to solve Problems 1 and 2 for general $\lambda$. In particular we generalize GREEDSAT leading to GENERALGREEDSAT, whose guarantee smoothly interpolates in terms of $\lambda$ between the bi-criterion factor by GREEDSAT in the case of $\lambda = 0$ and the constant factor of $1/2$ by the greedy algorithm in the case of $\lambda = 1$. For Problem 2 we generalize LOVÁSZ ROUND to obtain a relaxation algorithm (GENERALLOVÁSZ ROUND) that achieves an $m$-approximation for general $\lambda$. Motivated by the computational limitation of GENERALLOVÁSZ ROUND we also give a simple and efficient greedy heuristic called GENERALGREEDMIN that works for the homogeneous setting of Problem 2.

Lastly we demonstrate a number of applications of submodular partitioning in real-world machine learning problems. In particular we show Problem 1 is applicable in distributed training of statistical models. Problem 2, on the other hand, is useful for data clustering, image segmentation, and computational load balancing. In the experiments we empirically evaluate Problem 1 on data partitioning for ADMM and distributed deep neural network training. The efficacy of Problem 2 is tested on an unsupervised image segmentation task.

As an outline of this paper, we provide algorithms for SFA and SLB in Section 2. Algorithms for Problems 1 and 2 with general $\lambda$ are given in Section 3. Section 4 demonstrates several applications of Problems 1 and 2 to machine learning, and empirical validation of the proposed algorithms is given in Section 5. We conclude in Section 6.

## 2 Robust Submodular Partitioning (Problems 1 and 2 when $\lambda = 0$)

Notation: we define $f(j|S) \triangleq f(S \cup j) - f(S)$ as the gain of $j \in V$ in the context of $S \subseteq V$. Then, $f$ is submodular if and only if $f(j|S) \geq f(j|T)$ for all $S \subseteq T$ and $j \notin T$. Also, $f$ is monotone iff $f(j|S) \geq 0, \forall j \notin S, S \subseteq V$. We assume w.l.o.g. that the ground set is $V = \{1, 2, \cdots, n\}$.

### 2.1 Approximation Algorithms for SFA (Problem 1 with $\lambda = 0$)

We first investigate a special case of SFA with $m = 2$. When $m = 2$, the problem becomes

$$\max_{A \subseteq V} g(A), \tag{3}$$

where $g(A) = \min\{f_1(A), f_2(V \setminus A)\}$ and is submodular thanks to Theorem 2.1.

**Theorem 2.1.** *If $f_1$ and $f_2$ are monotone submodular, $\min\{f_1(A), f_2(V \setminus A)\}$ is also submodular.*

All proofs for the theoretical results are given in Appendix. Interestingly SFA for $m = 2$ can be equivalently formulated as unconstrained submodular maximization. This problem has been well studied in the literature [7, 11, 13]. A simple bi-directional randomized greedy algorithm [7] solves Eqn 3 with a tight factor of $1/2$. Applying this randomized algorithm to solve SFA then achieves a guarantee of $1/2$ matching the problem's hardness. However, the same idea does not apply to the general case of $m > 2$.

For general $m$, we approach SFA from the perspective of the greedy algorithms. Greedy is often the algorithm that practitioners use for combinatorial optimization problems since they are intuitive, simple to implement, and often lead to very good solutions. In this work we introduce two variants of a greedy algorithm – GREEDMAX (Alg. 1) and GREEDSAT (Alg. 2), suited to the homogeneous and heterogeneous settings, respectively.

---

**Algorithm 1:** GREEDMAX

---

1: Input: $f, m, V$.
2: Let $A_1 =, \ldots, = A_m = \emptyset; R = V$.
3: **while** $R \neq \emptyset$ **do**
4:     $j^* \in \operatorname{argmin}_j f(A_j)$;
5:     $a^* \in \operatorname{argmax}_{a \in R} f(a|A_{j^*})$
6:     $A_{j^*} \leftarrow A_{j^*} \cup \{a^*\}; R \leftarrow R \setminus a^*$
7: **end while**
8: Output $\{A_i\}_{i=1}^m$.

---

**GREEDMAX:** The key idea of GREEDMAX (see Alg. 1) is to greedily add an item with the maximum marginal gain to the block whose current solution is minimum. Initializing $\{A_i\}_{i=1}^m$ with the empty sets, the greedy flavor also comes from that it incrementally grows the solution by greedily improving the overall objective $\min_{i=1,\ldots,m} f_i(A_i)$ until $\{A_i\}_{i=1}^m$ forms a partition. Besides its simplicity, Theorem 2.2 offers the optimality guarantee.

**Theorem 2.2.** *Under the homogeneous setting ($f_i = f$ for all $i$),* GREEDMAX *is guaranteed to find a partition $\hat{\pi}$ such that*

$$\min_{i=1,\ldots,m} f(A_i^{\hat{\pi}}) \geq \frac{1}{m} \max_{\pi \in \Pi} \min_{i=1,\ldots,m} f(A_i^{\pi}). \tag{4}$$

By assuming the homogeneity of the $f_i$'s, we obtain a very simple $1/m$-approximation algorithm improving upon the state-of-the-art factor $1/(2m - 1)$ [27]. Thanks to the lazy evaluation trick as described in [34], Line 5 in Alg. 1 need not to recompute the marginal gain for every item in each round, leading GREEDMAX to scale to large data sets.

---

**Algorithm 2:** GREEDSAT

---

1: Input: $\epsilon, \{f_i\}_{i=1}^m, m, V, \alpha$.
2: Let $\bar{F}^c(\pi) = \frac{1}{m} \sum_{i=1}^m \min\{f_i(A_i^{\pi}), c\}$.
3: Let $c_{\min} = 0, c_{\max} = \min_i f_i(V)$
4: **while** $c_{\max} - c_{\min} \geq \epsilon$ **do**
5:     $c = \frac{1}{2}(c_{\max} + c_{\min})$
6:     $\hat{\pi}^c \in \operatorname{argmax}_{\pi \in \Pi} \bar{F}^c(\pi)$ // solved by GREEDSWP (Alg 3)
7:     **if** $\bar{F}^c(\hat{\pi}^c) < \alpha c$ **then**
8:         $c_{\max} = c$
9:     **else**
10:         $c_{\min} = c; \hat{\pi} \leftarrow \hat{\pi}^c$
11:     **end if**
12: **end while**
13: Output: $\hat{\pi}$.

---

**GREEDSAT:** Though simple and effective in the homogeneous setting, GREEDMAX performs arbitrarily poorly under the heterogeneous setting. Consider the following example: $V = \{v_1, v_2\}$, $f_1(v_1) = 1, f_1(v_2) = 0, f_1(\{v_1, v_2\}) = 1, f_2(v_1) = 1 + \epsilon, f_2(v_2) = 1, f_2(\{v_1, v_2\}) = 2 + \epsilon$. $f_1$

**Algorithm 3:** GREEDSWP

Input: $\{f_i\}_{i=1}^m$, $c$, $V$
Initialize: $A_1 = , \ldots , = A_m = \emptyset$, and $R \leftarrow V$
**while** $R \neq \emptyset$ **do**
  **for** $i = 1, \ldots, m$ **do**
    $\delta_i = \max_{r \in R} \min\{f_i(A_i \cup r), c\} - \min\{f_i(A_i), c\}$
    $a_i \in \operatorname{argmax}_{r \in R} \min\{f_i(A_i \cup r), c\} - \min\{f_i(A_i), c\}$
  $j \in \operatorname{argmax}_i \delta(i)$
  $A_j \leftarrow A_j \cup \{a_j\}$
  $R \leftarrow R \setminus a_j$
Output $\hat{\pi}^c = (A_1, \ldots, A_m)$.

and $f_2$ are monotone submodular. The optimal partition is to assign $v_1$ to $f_1$ and $v_2$ to $f_2$ leading to a solution of value 1. However, GREEDMAX may assign $v_1$ to $f_2$ and $v_2$ to $f_1$ leading to a solution of value 0. Therefore, GREEDMAX performs arbitrarily poorly on this example.

To this end we provide another algorithm – "Saturate" Greedy (GREEDSAT, see Alg. 2). The key idea of GREEDSAT is to relax SFA to a much simpler problem – Submodular Welfare (SWP), i.e., Problem 1 with $\lambda = 0$. Similar in flavor to the one proposed in [30] GREEDSAT defines an intermediate objective $\bar{F}^c(\pi) = \sum_{i=1}^m f_i^c(A_i^\pi)$, where $f_i^c(A) = \frac{1}{m} \min\{f_i(A), c\}$ (Line 2). The parameter $c$ controls the saturation in each block. It is easy to verify that $f_i^c$ satisfies submodularity for each $i$. Unlike SFA, the combinatorial optimization problem $\max_{\pi \in \Pi} \bar{F}^c(\pi)$ (Line 6) is much easier and is an instance of SWP. In this work, we solve Line 6 using the greedy algorithm as described in Alg 3, which attains a constant 1/2-approximation [14]. Moreover the lazy evaluation trick also applies for Alg 3 enabling the wrapper algorithm GREEDSAT scalable to large data sets. One can also use a more computationally expensive multi-linear relaxation algorithm as given in [44] to solve Line 6 with a tight factor $\alpha = (1 - 1/e)$. Setting the input argument $\alpha$ as the approximation factor for Line 6, the essential idea of GREEDSAT is to perform a binary search over the parameter $c$ to find the largest $c^*$ such that the returned solution $\hat{\pi}^{c^*}$ for the instance of SWP satisfies $\bar{F}^{c^*}(\hat{\pi}^{c^*}) \geq \alpha c^*$. GREEDSAT terminates after solving $O(\log(\frac{\min_i f_i(V)}{\epsilon}))$ instances of SWP. Theorem 2.3 gives a bi-criterion optimality guarantee.

**Theorem 2.3.** *Given $\epsilon > 0$, $0 \leq \alpha \leq 1$ and any $0 < \delta < \alpha$,* GREEDSAT *finds a partition such that at least $\lceil m(\alpha - \delta) \rceil$ blocks receive utility at least $\frac{\delta}{1 - \alpha + \delta}(\max_{\pi \in \Pi} \min_i f_i(A_i^\pi) - \epsilon)$.*

For any $0 < \delta < \alpha$ Theorem 2.3 ensures that the top $\lceil m(\alpha - \delta) \rceil$ valued blocks in the partition returned by GREEDSAT are $(\delta/(1 - \alpha + \delta) - \epsilon)$-optimal. $\delta$ controls the trade-off between the number of top valued blocks to bound and the performance guarantee attained for these blocks. The smaller $\delta$ is, the more top blocks are bounded, but with a weaker guarantee. We set the input argument $\alpha = 1/2$ (or $\alpha = 1 - 1/e$) as the worst-case performance guarantee for solving SWP so that the above theoretical analysis follows. However, the worst-case is often achieved only by very contrived submodular functions. For the ones used in practice, the greedy algorithm often leads to near-optimal solution ([30] and our own observations). Setting $\alpha$ as the actual performance guarantee for SWP (often very close to 1) can improve the empirical bound, and we, in practice, typically set $\alpha = 1$ to good effect.

**Algorithm 4:** MMAX

1: Input: $\{f_i\}_{i=1}^m$, $m$, $V$, partition $\pi^0$.
2: Let $t = 0$.
3: **repeat**
4:   **for** $i = 1, \ldots, m$ **do**
5:     Pick a subgradient $h_i$ at $A_i^{\pi^t}$ for $f_i$.
6:   **end for**
7:   $\pi^{t+1} \in \operatorname{argmax}_{\pi \in \Pi} \min_i h_i(A_i^\pi)$
8:   $t = t + 1$;
9: **until** $\pi^t = \pi^{t-1}$
10: Output: $\hat{\pi} \in \operatorname{argmax}_{\pi = \pi^1, \ldots, \pi^t} \min_i f_i(A_i^\pi)$.

**MMAX**: In parallel to GREEDSAT, we also introduce a semi-gradient based approach for solving SFA under the heterogeneous setting. We call this algorithm minorization-maximization (MMAX, see Alg. 4). Similar to the ones proposed in [25, 22, 20], the idea is to iteratively maximize tight lower bounds of the submodular functions. Submodular functions have tight modular lower bounds, which are related to the subdifferential $\partial_f(Y)$ of the submodular set function $f$ at a set $Y \subseteq V$, which is defined [15] as:

$$\partial_f(Y) = \{y \in \mathbb{R}^n : f(X) - y(X) \geq f(Y) - y(Y) \text{ for all } X \subseteq V\}. \tag{5}$$

For a vector $x \in \mathbb{R}^V$ and $X \subseteq V$, we write $x(X) = \sum_{j \in X} x(j)$. Denote a subgradient at $Y$ by $h_Y \in \partial_f(Y)$, the extreme points of $\partial_f(Y)$ may be computed via a greedy algorithm: Let $\sigma$ be a permutation of $V$ that assigns the elements in $Y$ to the first $|Y|$ positions ($\sigma(i) \in Y$ if and only if $i \leq |Y|$). Each such permutation defines a chain with elements $S_0^\sigma = \emptyset$, $S_i^\sigma = \{\sigma(1), \sigma(2), \ldots, \sigma(i)\}$, and $S_{|Y|}^\sigma = Y$. An extreme point $h_Y^\sigma$ of $\partial_f(Y)$ has each entry as

$$h_Y^\sigma(\sigma(i)) = f(S_i^\sigma) - f(S_{i-1}^\sigma). \tag{6}$$

Defined as above, $h_Y^\sigma$ forms a lower bound of $f$, tight at $Y$ — i.e., $h_Y^\sigma(X) = \sum_{j \in X} h_Y^\sigma(j) \leq f(X), \forall X \subseteq V$ and $h_Y^\sigma(Y) = f(Y)$. The idea of MMAX is to consider a modular lower bound tight at the set corresponding to each block of a partition. In other words, at iteration $t + 1$, for each block $i$, we approximate $f_i$ with its modular lower bound tight at $A_i^{\pi^t}$ and solve a modular version of Problem 1 (Line 7), which admits efficient approximation algorithms [2]. MMAX is initialized with a partition $\pi^0$, which is obtained by solving Problem 1, where each $f_i$ is replaced with a simple modular function $f_i'(A) = \sum_{a \in A} f_i(a)$. The following worst-case bound holds:

**Theorem 2.4.** MMAX *achieves a worst-case guarantee of*

$$O(\min_i \frac{1 + (|A_i^{\hat{\pi}}| - 1)(1 - \kappa_{f_i}(A_i^{\hat{\pi}}))}{|A_i^{\hat{\pi}}|\sqrt{m}\log^3 m}),$$

*where* $\hat{\pi} = (A_1^{\hat{\pi}}, \cdots, A_m^{\hat{\pi}})$ *is the partition obtained by the algorithm, and*

$$\kappa_f(A) = 1 - \min_{v \in V} \frac{f(v|A \setminus v)}{f(v)} \in [0, 1]$$

*is the curvature of a submodular function* $f$ *at* $A \subseteq V$.

When each submodular function $f_i$ is modular, i.e., $\kappa_{f_i}(A) = 0, \forall A \subseteq V, i$, the approximation guarantee of MMAX becomes $O(\frac{1}{\sqrt{m}\log^3 m})$, which matches the performance of the approximation algorithm for the modular problem. When each $f_i$ is fully curved, i.e., $\kappa_{f_i} = 1$, we still obtain a bounded guarantee of $O(\frac{1}{n\sqrt{m}\log^3 m})$. Theorem 2.4 suggests that the performance of MMAX improves as the curvature $\kappa_{f_i}$ of each objective $f_i$ decreases. This is natural since MMAX essentially uses the modular lower bounds as the proxy for each objective and optimizes with respect to the proxy functions. Lower the curvature of the objectives, the better the modular lower bounds approximate, hence better performance guarantee.

Since the modular version of SFA is also NP-hard and cannot be exactly solved in polynomial time, we cannot guarantee that successive iterations of MMAX improves upon the overall objective. However we still obtain the following Theorem giving a bounded performance gap between the successive iterations.

**Theorem 2.5.** *Suppose modular version of SFA is solved with an approximation factor* $\alpha \leq 1$, *we have for each iteration* $t$ *that*

$$\min_i f_i(A_i^{\pi_t}) \geq \alpha \min_i f_i(A_i^{\pi_{t-1}}). \tag{7}$$

## 2.2 Approximation Algorithms for SLB (Problem 2 with $\lambda = 0$)

In this section, we investigate the problem of submodular load balancing (SLB). It is a special case of Problem 2 with $\lambda = 0$. We first analyze the hardness of SLB. We then show a Lovász extension-based algorithm with a guarantee matching the problem's hardness. Lastly we describe a more efficient supergradient based algorithm.

Existing hardness for SLB is shown to be $o(\sqrt{n/\log n})$ [40]. However it is independent of $m$, and [40] assumes $m = \Theta(\sqrt{n/\log n})$ in their analysis. In most of the applications of SLB (cf. Section 4), we find that the parameter $m$ is such that $m \ll n$ and can sometimes be treated as a

constant w.r.t. $n$. To this end we offer a more general hardness analysis that is dependent directly on $m$.

**Theorem 2.6.** *For any $\epsilon > 0$, SLB cannot be approximated to a factor of $(1 - \epsilon)m$ for any $m = o(\sqrt{n/\log n})$ with polynomial number of queries even under the homogeneous setting.*

Though the proof technique for Theorem 2.6 mostly carries over from [40], the result strictly generalizes the analysis in [40]. For any choice of $m = o(\sqrt{n/\log n})$ Theorem 2.6 implies that it is information theoretically hard to approximate SLB better than $m$ even for the homogeneous setting. For the rest of the paper, we assume $m = o(\sqrt{n/\log n})$ for SLB, unless stated otherwise. It is worth pointing out that arbitrary partition $\pi \in \Pi$ already achieves the best approximation factor of $m$ that one can hope for under the homogeneous setting. Denote $\pi^*$ as the optimal partitioning for SLB, i.e., $\pi^* \in \operatorname{argmin}_{\pi \in \Pi} \max_i f(A_i^\pi)$. This can be verified by considering the following:

$$\max_i f(A_i^\pi) \leq f(V) \leq \sum_{i=1}^m f(A_i^{\pi^*}) \leq m \max_i f(A_i^{\pi^*}). \tag{8}$$

It is therefore theoretically interesting to consider only the heterogeneous setting.

---

**Algorithm 5:** LOVÁSZ ROUND

---
1: Input: $\{f_i\}_{i=1}^m, \{\tilde{f}_i\}_{i=1}^m, m, V$.
2: Solve for $\{x_i^*\}_{i=1}^m$ via convex relaxation.
3: Rounding: Let $A_1 =, \ldots, = A_m = \emptyset$.
4: **for** $j = 1, \ldots, n$ **do**
5:    $\hat{i} \in \operatorname{argmax}_i x_i^*(j)$; $A_{\hat{i}} = A_{\hat{i}} \cup j$
6: **end for**
7: Output $\hat{\pi} = \{A_i\}_{i=1}^m$.

---

**LOVÁSZ ROUND:** Next we propose a tight algorithm – LOVÁSZ ROUND (see Alg. 5) for the heterogeneous setting of SLB. The algorithm proceeds as follows: (1) apply the Lovász extension of submodular functions to relax SLB to a convex program, which is exactly solved to a fractional solution (Line 2); (2) map the fractional solution to a partition using the $\theta$-rounding technique as proposed in [23] (Line 3 - 6). The Lovász extension, which naturally connects a submodular function $f$ with its convex relaxation $\tilde{f}$, is defined as follows: given any $x \in [0, 1]^n$, we obtain a permutation $\sigma_x$ by ordering its elements in non-increasing order, and thereby a chain of sets $S_0^{\sigma_x} \subset, \ldots, \subset S_n^{\sigma_x}$ with $S_j^{\sigma_x} = \{\sigma_x(1), \ldots, \sigma_x(j)\}$ for $j = 1, \ldots, n$. The Lovász extension $\tilde{f}$ for $f$ is the weighted sum of the ordered entries of $x$:

$$\tilde{f}(x) = \sum_{j=1}^n x(\sigma_x(j))(f(S_j^{\sigma_x}) - f(S_{j-1}^{\sigma_x})). \tag{9}$$

Given the convexity of the $\tilde{f}_i$'s , SLB is relaxed to the following convex program:

$$\min_{x_1, \ldots, x_m \in [0,1]^n} \max_i \tilde{f}_i(x_i), \text{ s.t } \sum_{i=1}^m x_i(j) \geq 1, \text{ for } j = 1, \ldots, n \tag{10}$$

Denoting the optimal solution for Eqn 10 as $\{x_1^*, \ldots, x_m^*\}$, the $\theta$-rounding step simply maps each item $j \in V$ to a block $\hat{i}$ such that $\hat{i} \in \operatorname{argmax}_i x_i^*(j)$ (ties broken arbitrarily). The bound for LOVÁSZ ROUND is as follows:

**Theorem 2.7.** LOVÁSZ ROUND *is guaranteed to find a partition $\hat{\pi} \in \Pi$ such that*

$$\max_i f_i(A_i^{\hat{\pi}}) \leq m \min_{\pi \in \Pi} \max_i f_i(A_i^\pi)$$

.

We remark that, to the best of our knowledge, LOVÁSZROUND is the first algorithm that is tight and that gives an approximation in terms of $m$ for the heterogeneous setting.

**MMIN**: Similar to MMAX for SFA, we propose Majorization-Minimization (MMIN, see Alg. 6) for SLB. Here, we iteratively choose modular upper bounds, which are defined via superdifferentials $\partial^f(Y)$ of a submodular function [26] at $Y$:

$$\partial^f(Y) = \{y \in \mathbb{R}^n : f(X) - y(X) \leq f(Y) - y(Y); \text{for all } X \subseteq V\}. \tag{11}$$

---

**Algorithm 6:** MMIN

1: Input: $\{f_i\}_{i=1}^m$, $m$, $V$, partition $\pi^0$.
2: Let $t = 0$
3: **repeat**
4:     **for** $i = 1, \ldots, m$ **do**
5:         Pick a supergradient $m_i$ at $A_i^{\pi^t}$ for $f_i$.
6:     **end for**
7:     $\pi^{t+1} \in \operatorname{argmin}_{\pi \in \Pi} \max_i m_i(A_i^\pi)$
8:     $t = t + 1$;
9: **until** $\pi^t = \pi^{t-1}$
10: Output: $\pi^t$.

---

Moreover, there are specific supergradients [21, 25] that define the following two modular upper bounds (when referring to either one, we use $m_X^f$):

$$m_{X,1}^f(Y) \triangleq f(X) - \sum_{j \in X \setminus Y} f(j | X \setminus j) + \sum_{j \in Y \setminus X} f(j | \emptyset),$$

$$m_{X,2}^f(Y) \triangleq f(X) - \sum_{j \in X \setminus Y} f(j | V \setminus j) + \sum_{j \in Y \setminus X} f(j | X).$$

Then $m_{X,1}^f(Y) \geq f(Y)$ and $m_{X,2}^f(Y) \geq f(Y), \forall Y \subseteq V$ and $m_{X,1}^f(X) = m_{X,2}^f(X) = f(X)$. At iteration $t + 1$, for each block $i$, MMIN replaces $f_i$ with a choice of its modular upper bound $m_i$ tight at $A_i^{\pi^t}$ and solves a modular version of Problem 2 (Line 7), for which there exists an efficient LP relaxation based algorithm [32]. Similar to MMAX, the initial partition $\pi^0$ is obtained by solving Problem 2, where each $f_i$ is substituted with $f_i'(A) = \sum_{a \in A} f_i(a)$. The following worst-case bound holds:

**Theorem 2.8.** MMIN *achieves a worst-case guarantee of* $(2 \max_i \frac{|A_i^{\pi^*}|}{1 + (|A_i^{\pi^*}| - 1)(1 - \kappa_{f_i}(A_i^{\pi^*}))})$, *where* $\pi^* = (A_1^{\pi^*}, \cdots, A_m^{\pi^*})$ *denotes the optimal partition.*

Similar to MMax, we can show that MMin has bounded performance gaps in successive iterations.

**Theorem 2.9.** *Suppose the modular version of SLB can be solved with an approximation factor* $\alpha \geq 1$, *we have for each iteration* $t$ *that*

$$\max_i f_i(A_i^{\pi^t}) \leq \alpha \max_i f_i(A_i^{\pi^{t-1}}). \tag{12}$$

## 3   General Submodular Partitioning (Problems 1 and 2 when $0 < \lambda < 1$)

In this section we study Problem 1 and Problem 2, in the most general case, i.e., $0 < \lambda < 1$. We use the proposed algorithms for the special cases of Problems 1 and 2 as the building blocks to design algorithms for the general scenarios ($0 < \lambda < 1$). We first propose a simple and generic scheme that provides performance guarantee in terms of $\lambda$ for both problems. We then generalize the proposed GREEDSAT to obtain a more practically interesting algorithm for Problem 1. For Problem 2 we generalize LOVÁSZ ROUND to obtain a relaxation based algorithm.

**Extremal Combination Scheme:** First we describe the scheme that works for both problem 1 and 2. It naturally combines an algorithm for solving the worst-case problem ($\lambda = 0$) with an algorithm for solving the average case ($\lambda = 1$). We use Problem 1 as an example, but the same scheme easily works for Problem 2. Denote ALGWC as the algorithm for the worst-case problem (i.e. SFA), and ALGAC as the algorithm for the average case (i.e., SWP). The scheme is to first obtain a partition $\hat{\pi}_1$ by running ALGWC on the instance of Problem 1 with $\lambda = 0$ and a second partition $\hat{\pi}_2$ by running ALGAC with $\lambda = 1$. Then we output one of $\hat{\pi}_1$ and $\hat{\pi}_2$, with which the higher valuation for Problem 1 is achieved. We call this scheme COMBSFASWP. Suppose ALGWC solves the worst-case problem with a factor $\alpha \leq 1$ and ALGAC for the average case with $\beta \leq 1$. When applied to Problem 2 we refer to this scheme as COMBSLBSMP ($\alpha \geq 1$ and $\beta \geq 1$). The following guarantee holds for both schemes:

**Theorem 3.1.** *For any* $\lambda \in (0, 1)$ COMBSFASWP *solves Problem 1 with a factor* $\max\{\frac{\beta\alpha}{\lambda\beta+\alpha}, \lambda\beta\}$ *in the* heterogeneous *case, and* $\max\{\min\{\alpha, \frac{1}{m}\}, \frac{\beta\alpha}{\lambda\beta+\alpha}, \lambda\beta\}$ *in the* homogeneous *case. Similarly,*

COMBSLBSMP *solves Problem* 2 *with a factor* $\min\{\frac{m\alpha}{m\bar{\lambda}+\lambda}, \beta(m\bar{\lambda}+\lambda)\}$ *in the* heterogeneous *case, and* $\min\{m, \frac{m\alpha}{m\bar{\lambda}+\lambda}, \beta(m\bar{\lambda}+\lambda)\}$ *in the* homogeneous *case.*

---

**Algorithm 7:** GENERALGREEDSAT

---

1: Input: $\epsilon, \{f_i\}_{i=1}^m, m, V, \lambda, \alpha$.
2: Let $\bar{F}_\lambda^c(\pi) = \frac{1}{m} \sum_{i=1}^m \min\{\bar{\lambda} f_i(A_i^\pi) + \lambda \frac{1}{m} \sum_{j=1}^m f_j(A_j^\pi), c\}$.
3: Let $c_{\min} = 0, c_{\max} = \sum_{i=1}^m f_i(V)$
4: **while** $c_{\max} - c_{\min} \geq \epsilon$ **do**
5:     $c = \frac{1}{2}(c_{\max} + c_{\min})$
6:     $\hat{\pi}^c \in \text{argmax}_{\pi \in \Pi} \bar{F}_\lambda^c(\pi)$ // solved by GREEDSWP (Alg. 3)
7:     **if** $\bar{F}^c(\hat{\pi}^c) < \alpha c$ **then**
8:         $c_{\max} = c$
9:     **else**
10:        $c_{\min} = c; \hat{\pi} \leftarrow \hat{\pi}^c$
11:     **end if**
12: **end while**
13: Output: $\hat{\pi}$.

---

**GeneralGreedSat:** The drawback of COMBSFASWP and COMBSLBSMP is that they do not explicitly exploit the trade-off between the average-case and worst-case in terms of $\lambda$. To obtain more practically interesting algorithms, we first give GENERALGREEDSAT (See Alg. 7) that generalizes GREEDSAT to solve Problem 1 for general $\lambda$. The key idea of GENERALGREEDSAT is again to relax Problem 1 to a simpler submodular welfare problem (SWP). Similar to GREEDSAT we define an intermediate objective:

$$\bar{F}_\lambda^c(\pi) = \frac{1}{m} \sum_{i=1}^m \min\{\bar{\lambda} f_i(A_i^\pi) + \lambda \frac{1}{m} \sum_{j=1}^m f_j(A_j^\pi), c\}. \quad (13)$$

It is easy to verify that the combinatorial optimization problem $\max_{\pi \in \Pi} \bar{F}_\lambda^c(\pi)$ (Line 6) can be formulated as the submodular welfare problem, for which we can solve efficiently with GREEDSWP (see Alg. 3). Defining $\alpha$ as the optimality guarantee of the algorithm for solving Line 6 GENERAL-GREEDSAT solves Problem 1 with the following bounds:

**Theorem 3.2.** *Given* $\epsilon > 0$, $0 \leq \alpha \leq 1$, *and* $0 \leq \lambda \leq 1$, GENERALGREEDSAT *finds a partition* $\hat{\pi}$ *that satisfies the following:*

$$\bar{\lambda} \min_i f_i(A_i^{\hat{\pi}}) + \lambda \frac{1}{m} \sum_{i=1}^m f_i(A_i^{\hat{\pi}}) \geq \lambda \alpha (OPT - \epsilon) \quad (14)$$

*where* $OPT = \max_{\pi \in \Pi} \bar{\lambda} \min_i f_i(A_i^\pi) + \lambda \frac{1}{m} \sum_{i=1}^m f_i(A_i^\pi)$.

*Moreover, let* $F_{\lambda,i}(\pi) = \bar{\lambda} f_i(A_i^\pi) + \lambda \frac{1}{m} \sum_{j=1}^m f_j(A_j^\pi)$. *Given any* $0 < \delta < \alpha$, *there is a set* $I \subseteq \{1, \ldots, m\}$ *such that* $|I| \geq \lceil m(\alpha - \delta) \rceil$ *and*

$$F_{i,\lambda}(\hat{\pi}) \geq \max\{\frac{\delta}{1-\alpha+\delta}, \lambda\alpha\}(OPT - \epsilon), \forall i \in I. \quad (15)$$

Eqn 15 in Theorem 3.2 reduces to Theorem 2.3 when $\lambda = 0$, i.e., it recovers the bi-criterion guarantee in Theorem 2.3 for the worst-case scenario ($\lambda = 0$). Eqn 14 in Theorem 3.2 implies that $\alpha$-approximation for the average-case objective can almost be recovered by GENERALGREEDSAT if $\lambda = 1$. Moreover Theorem 3.2 shows that the guarantee of GENERALGREEDSAT improves as $\lambda$ increases suggesting that Problem 1 becomes easier as the mixed objective weights more on the average-case objective. We also point out that the approximation guarantee of GENERALGREEDSAT smoothly interpolates the two extreme cases in terms of $\lambda$.

**GeneralLovász Round:** Next we focus on designing practically more interesting algorithms for Problem 2 with general $\lambda$. In particular we generalize LOVÁSZROUND leading to GENERALLOVÁSZ ROUND as shown in Alg. 8. Sharing the same idea with LOVÁSZROUND, GENERALLOVÁSZROUND first relaxes Problem 2 as a convex program (defined in Eqn 16) using the Lovász extension of each submodular objective. Given the fractional solution to the convex program $\{x_i^*\}_{i=1}^m$, the algorithm

---
**Algorithm 8:** GENERALLOVÁSZROUND
---
1: Input: $\{f_i\}_{i=1}^m$, $\{\tilde{f}_i\}_{i=1}^m$, $\lambda$, $m$, $V$.
2: Solve

$$\min_{x_1,\ldots,x_m \in [0,1]^n} \max_i \bar{\lambda}\tilde{f}_i(x_i) + \lambda\frac{1}{m}\sum_{j=1}^m \tilde{f}_j(x_j), \text{ s.t } \sum_{i=1}^m x_i(j) \geq 1, \text{ for } j = 1,\ldots,n \quad (16)$$

for $\{x_i^*\}_{i=1}^m$ via convex relaxation.
3: Rounding: Let $A_1 =,\ldots,= A_m = \emptyset$.
4: **for** $j = 1,\ldots,n$ **do**
5: $\quad \hat{i} \in \text{argmax}_i \, x_i^*(j); A_{\hat{i}} = A_{\hat{i}} \cup j$
6: **end for**
7: Output $\hat{\pi} = \{A_i\}_{i=1}^m$.
---

then rounds it to a partition using the $\theta$-rounding technique (Line 3- 6). Note the rounding technique used for GENERALLOVÁSZROUND is the same as in LOVÁSZROUND. The following Theorem holds:

**Theorem 3.3.** GENERALLOVÁSZROUND *is guaranteed to find a partition* $\hat{\pi} \in \Pi$ *such that*

$$\max_i \bar{\lambda}f_i(A_i^{\hat{\pi}}) + \lambda\frac{1}{m}\sum_{j=1}^m f_j(A_j^{\hat{\pi}}) \leq m \min_{\pi \in \Pi} \max_i \bar{\lambda}f_i(A_i^{\pi}) + \lambda\frac{1}{m}\sum_{j=1}^m f_j(A_j^{\pi}). \quad (17)$$

Theorem 3.3 generalizes Theorem 2.7 when $\lambda = 0$. Moreover we achieve a factor of $m$ by GENERALLOVÁSZ ROUND for any $\lambda$. Though the approximation guarantee is independent of $\lambda$ the algorithm naturally exploits the trade-off between the worst-case and average-case objectives in terms of $\lambda$. The drawback of GENERALLOVÁSZROUND is that it requires high order polynomial queries of the Lovász extension of the submodular objectives, hence is not computationally feasible for even medium sized tasks. Moreover, if we restrict ourselves to the homogeneous setting ($f_i$'s are identical), it is easy to verify that arbitrary partitioning already achieves a guarantee of $m$ while Problem 2, in general, cannot be approximated better than $m$ as shown in Theorem 2.6.

**GeneralGreedMin:** In this case, we should resort to intuitive heuristics that are scalable to large-scale applications to solve Problem 2 with general $\lambda$. To this end we design a greedy heuristic called GENERALGREEDMIN (see Alg. 9).

---
**Algorithm 9:** GENERALGREEDMIN
---
1: Input: $f$, $m$, $V$, $0 \leq \lambda \leq 1$;
2: Solve $S_{\text{seed}} \in \text{argmax}_{S \subseteq V; |S|=m} f(S)$ for $m$ seeds with $S_{\text{seed}} = \{s_1,\ldots,s_m\}$.
3: Initialize each block $i$ by the seeds as $A_i \leftarrow \{s_i\}$, $\forall i$.
4: Initialize a counter as $k = m$ and $R = V \setminus S_{\text{seed}}$.
5: **while** $R \neq \emptyset$ **do**
6: $\quad$ **if** $k \leq (1-\lambda)|V|$ **then**
7: $\quad\quad j^* \in \text{argmin}_j f(A_j)$
8: $\quad\quad a^* \in \min_{a \in R} f(a|A_{j^*})$
9: $\quad\quad A_{j^*} \leftarrow A_{j^*} \cup a^*; R \leftarrow R \setminus a^*$
10: $\quad$ **else**
11: $\quad\quad$ **for** $i = 1,\ldots,m$ **do**
12: $\quad\quad\quad a_i^* \in \text{argmin}_{a \in R} f(a|A_i)$
13: $\quad\quad$ **end for**
14: $\quad\quad j^* \in \text{argmin}_{i=1,\ldots,m} f(a_i^*|A_i)$;
15: $\quad\quad A_{j^*} \leftarrow A_{j^*} \cup a^*; R \leftarrow R \setminus a_{j^*}^*$
16: $\quad$ **end if**
17: $\quad k = k + 1$;
18: **end while**
19: Output $\{A_i\}_{i=1}^m$.
---

The algorithm first solves a constrained submodular maximization on $f$ to obtain a set $S_{\text{seed}}$ of $m$ seeds. Since $f$ is submodular, maximizing $f$ always leads to a set of diverse seeds, where the diversity is measured by the objective $f$. We initialize each block $A_i$ with one seed from $S_{\text{seed}}$. Defining $k$ as the number of items that have already been assigned. The main algorithm consists of two phases. In the first phase ($k \leq (1 - \lambda)|V|$), we, for each iteration, assign the item that has the smallest marginal gain to the block whose valuation is the smallest. Since the functions are all monotone, any additions to a block can (if anything) only increase its value. Such procedure inherently minimizes the worst-case objective, since it chooses the minimum valuation block to add to in order to keep the maximum valuation block from growing further. In the second phase ($k > \lambda|V|$), we assign an item such that its marginal gain is the smallest among all remaining items and all blocks. The greedy procedure in this phase, on the hand, is suitable for minimizing the average-case objective, since it, in each iteration, assigns an item so that the valuation of the average-case objective increases the least. The trade-off between the worse-case and the average-case objectives is controlled by $\lambda$, which is used as the input argument to the algorithm. In particular, $\lambda$ controls the fraction of the iterations in the algorithm to optimize the average-case objective. When $\lambda = 1$, the algorithm solely focuses on the average-case objective, while only the worst-case objective is minimized if $\lambda = 0$.

In general GENERALGREEDMIN requires $O(m|V|^2)$ function valuations, which may still be computationally difficult for large-scale applications. In practice, one can relax the condition in Line 8 and 12. Instead of searching among all items in $R$, one can, in each round, randomly select a subset $\hat{R} \subseteq R$ and choose an item with the smallest marginal gain from only the subset $\hat{R}$. The resultant computational complexity is reduced to $O(m|\hat{R}||V|)$ function valuations. Empirically we observe that GENERALGREEDMIN can be sped up more than 100 times by this trick without much performance loss.

## 4 Applications of Problems 1 and 2 in Machine Learning and Data Science

In this section we show a number of applications of submodular partitioning in machine learning and data science. We first discuss a number of distributed statistical training applications for which Problem 1 can be useful to obtain good partitioning of the data set. Next we demonstrate how data clustering, image segmentation, and computational load balancing can be formulated as an instance of Problem 2.

### 4.1 Applications of Problem 1

**Distributed statistical training:** An important machine learning application is distributed training of statistical models. As data set sizes grow, the need for statistical training procedures tolerant of the distributed data partitioning becomes more important. Existing schemes are often developed and performed assuming data samples are distributed to their computational clients in an arbitrary or random fashion. As an alternate strategy, if the data is intelligently partitioned such that each block of samples can itself lead to a good approximate solution, a consensus amongst the distributed results could be reached more quickly than when under a poor partitioning. Submodular functions can in fact express the value of a subset of training data for certain machine learning risk functions, e.g., [46] in the case of classification. Using these functions within Problem 1, one can expect a partitioning (by formulating the problem as an instance of Problem 1, $\lambda \approx 0$) where each block is a good *representative* of the entire set, thereby achieving faster convergence in distributed settings. We demonstrate empirically, in Section 5, that this provides better results on several machine learning tasks, including the training of deep neural networks.

### 4.2 Applications of Problem 2

**Data clustering and image segmentation:** Submodular functions naturally capture notions of interacting cooperative costs and homogeneity and thus are useful for clustering and image segmentation [36, 5, 29, 28]. While the average case instance (Problem 2 with $\lambda = 1$) has been used before, a more worst-case variant (i.e., Problem 2 with $\lambda \approx 0$) is useful to produce balanced clusterings (i.e., the submodular valuations of all the blocks should be similar to each other). Problem 2 also addresses a problem in image segmentation, namely how to use only submodular functions (which are instances of pseudo-Boolean functions) for multi-label (i.e., non-Boolean) image segmentation.

Problem 2 addresses this problem by allowing each segment $j$ to have its own submodular function $f_j$, and the objective measures the homogeneity $f_j(A_j^\pi)$ of segment $j$ based on the image pixels $A_j^\pi$ assigned to it. Moreover, by combining the average case and the worst case objectives, one can achieve a tradeoff between the two. Empirically, we evaluate our algorithms on unsupervised image segmentation (Section 5) and find that it outperforms other clustering methods including $k$-means, $k$-medoids, graph cut, and spectral clustering.

**Computational load balancing:** Submodularity also accurately represents computational costs in distributed systems, as shown in [33]. They consider a problem of text data partitioning for balancing memory demands. Given a large collection of documents $V = \{v_1, \ldots, v_n\}$, the goal is to distribute the documents into $m$ machines such that the memory requirements across the machines are balanced and minimized. Each document $v \in V$ consists a set of keys, and let $U = \{u_1, \ldots, u_k\}$ be the set of all possible keys. $|U|$ can be in the order of billions (e.g., the set of all unigrams, bigrams, and trigrams). Let $N(v_i) \subseteq U$ be the set of keys contained by the document $v_i$. Given a partition $\pi = (A_1^\pi, \ldots, A_m^\pi)$ of the documents $V$, the number of unique keys associated with the collection $A_i^\pi$ on machine $i$ is expressed as

$$f_{\text{sc}}(A_i^\pi) = |\cup_{v \in A_i^\pi} N(v)|, \tag{18}$$

where $f_{\text{sc}}$ is the set cover function. A hard constraint for a partition $\{X_1, \ldots, X_m\}$ to satisfy is that the number of unique keys on each machine has to be small enough so that they can be cached into each machine's memory. Since the memory needed to cache the keys on machine $i$ is proportional to $|\cup_{v \in X_i} N(v)|$, a feasible partition of the documents satisfying the memory requirement can, therefore, be found by solving the following:

$$\min_{\pi \in \Pi} \max_{i=1,\ldots,m} f_{\text{sc}}(A_i^\pi), \tag{19}$$

which is an instance of Problem 2 for $\lambda = 0$ with $f_{\text{sc}}$ as the objective function.

## 5  Experiments

In this section we empirically evaluate the algorithms proposed for Problems 1 and 2. We first compare the performance of the various algorithms discussed in this paper on a synthetic data set. We then evaluate some of the scalable algorithms proposed for Problems 1 and 2 on large-scale real-world data partitioning applications including distributed ADMM, distributed neural network training, and lastly unsupervised image segmentation tasks.

### 5.1  Experiments on Synthetic Data

In this section we evaluate separately on four different cases: Problem 1 with $\lambda = 0$ (SFA), Problem 2 with $\lambda = 0$ (SLB), Problem 1 with $0 < \lambda < 1$, and Problem 2 with $0 < \lambda < 1$. Since some of the algorithms, such as the Ellipsoidal Approximations [17] and Lovász relaxation algorithms, are computationally intensive, we restrict ourselves to only 40 data instances, i.e., $|V| = 40$. For simplicity we only evaluate on the homogeneous setting ($f_i$'s are identical). For each case we test with two types of submodular functions: facility location function, and the set cover function. The facility location function is defined as follows:

$$f_{\text{fac}}(A) = \sum_{v \in V} \max_{a \in A} s_{v,a}, \tag{20}$$

where $s_{v,a}$ is the similarity between item $v$ and $a$ and is symmetric, i.e., $s_{v,a} = s_{a,v}$ for any pair of $v$ and $a$. We define $f_{\text{fac}}$ on a complete similarity graph with each edge weight $s_{v,a}$ sampled uniformly and independently from $[0, 1]$. The set cover function $f_{\text{sc}}$ is as defined in Eqn 18, where we choose $|U| = 40$. The set cover function is defined by a bipartite graph between $V$ and $U$, where we define an edge between an item $v \in V$ and a key $u \in U$ independently with probability $p = 0.2$.

**Problem 1**  For $\lambda = 0$, i.e., SFA, we compare among 6 algorithms: GREEDMAX, GREEDSAT, MMAX, Balanced Partition (BP), Ellipsoid Approximation (EA) [17], and Binary Search algorithm (BS) [27]. Balanced Partition method simply partitions the ground set $V$ into $m$ blocks such that the size of each block is balanced and is either $\lceil \frac{|V|}{m} \rceil$ or $\lfloor \frac{|V|}{m} \rfloor$. We run 100 randomly generated instances of the balanced partition method. GREEDSAT is implemented with the choice of the hyperparameter $\alpha = 1$. We compare the performance of these algorithms in Figure 1a and 1b, where we vary the number of blocks $m$ from 2 to 14. The three proposed algorithms (GREEDMAX, GREEDSAT, and

(a) Problem 1 on $f_{\text{fac}}$ with $\lambda = 0$  (b) Problem 1 on $f_{\text{sc}}$ with $\lambda = 0$

(c) Problem 1 on $f_{\text{fac}}$ with varying $\lambda$  (d) Problem 1 on $f_{\text{sc}}$ with varying $\lambda$

Figure 1: Synthetic experiments on Problem 1 with $\lambda = 0$ on facility location function (a) and set cover function (b). Problem 1 with general $0 < \lambda < 1$ on facility location function (c) and set cover function (d).

MMAX) significantly and consistently outperform all baseline methods for both $f_{\text{fac}}$ and $f_{\text{sc}}$. Among the proposed algorithms we observe that GREEDMAX, in general, yields the superior performance. Given the empirical success, computational efficiency, and tight theoretical guarantee, we suggest GREEDMAX as the first choice of algorithm to solve SFA under the homogeneous setting.

Next we evaluate Problem 1 with general $0 < \lambda < 1$. Baseline algorithms for SFA such as Ellipsoidal Approximations, Binary Search do not apply to the mixed scenario. Similarly the proposed algorithms such as GREEDMAX, MMAX do not simply generalize to this scenario. We therefore only compare GENERALGREEDSAT with the Balanced Partition as a baseline. The results are summarized in Figure 1c and 1d. We observe that GENERALGREEDSAT consistently and significantly outperform even the best of 100 instances of the baseline method for all cases of $\lambda$.

**Problem 2** For $\lambda = 0$, i.e., SLB, we compare among 5 algorithms: LOVÁSZROUND, MMIN, GEN-ERALGREEDMIN, Ellipsoid Approximation (EA) [17], and Balanced Partition [41]. We implement GENERALGREEDMIN with the input argument $\lambda = 0$. We also run 100 randomly generated instances of the Balanced Partition method as a baseline. We show the results in Figure 2a and 2b. Among all five algorithms MMIN and GENERALGREEDMIN, in general, perform the best. Between MMIN and GENERALGREEDMIN we observe that GENERALGREEDMIN performs marginally better, especially on $f_{\text{sc}}$. The computationally intensive algorithms, such as Ellipsoid Approximation and LOVÁSZ ROUND, do not perform well, though they carry better worst-case approximation factors for the heterogeneous setting.

Lastly we evaluate Problem 2 with general $0 < \lambda < 1$. Since MMIN and Ellipsoid Approximation do not apply for the mixed scenario, we test only on GENERALLOVÁSZROUND, GENERALGREEDMIN, and Balanced Partition. Again we test on 100 instances of randomly generated balanced partitions. We vary $\lambda$ in this experiment. The results are shown in Figure 2c and 2d. The best performance is consistently achieved by GENERALGREEDMIN.

(a) Problem 2 on $f_{\text{fac}}$ with $\lambda = 0$   (b) Problem 2 on $f_{\text{sc}}$ with $\lambda = 0$

(c) Problem 2 on $f_{\text{fac}}$ with varying $\lambda$   (d) Problem 2 on $f_{\text{sc}}$ with varying $\lambda$

Figure 2: Synthetic experiments on Problem 2 with $\lambda = 0$ on facility location function (a) and set cover function (b). Problem 2 with general $0 < \lambda < 1$ on facility location function (c) and set cover function (d).

## 5.2   Problem 1 for Distributed Training

In this section we focus on applications of Problem 1 to real-wold machine learning problems. In particular we examine how a partition obtained by solving Problem 1 with certain instances of submodular functions perform for distributed training of various statistical models.

**Distributed Convex Optimization:**   We first consider data partitioning for distributed convex optimization. We evaluate the distributed convex optimization on a text categorization task. We use 20 Newsgroup data set [4], which consists of 18,774 articles divided almost evenly across 20 classes. The text categorization task is to classify an article into one newsgroup (of twenty) to which it was posted. We randomly split 2/3 and 1/3 of the whole data as the training and test data. The task is solved as a multi-class classification problem, which we formulate as an $\ell_2$ regularized logistic regression. We solve this convex optimization problem in a distributive fashion, where the data samples are partitioned and distributed across multiple machines. In particular we implement an ADMM algorithm as described in [4] to solve the distributed convex optimization problem. Given a partition $\pi$ of the training data into $m$ separate clients, in each iteration, ADMM first assigns each client $i$ to solve an $\ell_2$ regularized logistic regression on its block of data $A_i^\pi$ using L-BFGS, aggregate the solutions from all $m$ clients according to the ADMM update rules, and then sends the aggregated solution back to each client. This iterative procedure is carried out so that solutions on all clients converge to a consensus, which is the global solution of the overall large-scale convex optimization problem.

We formulate the data partitioning problem as an instance of SFA (Problem 1 with $\lambda = 0$) under the homogeneous setting. In the experiment, we solve the data partitioning using GREEDMAX, since it is efficient and attains the tightest guarantee among all algorithms proposed for this setting. Note, however, GREEDSAT and MMAX may also be used in the experiment. We model the utility of a data

subset using the feature-based submodular function [47, 46, 42], which has the form:

$$f_{\text{fea}}(A) = \sum_{u \in \mathcal{U}} m_u(V) \log m_u(A),  \tag{21}$$

where $\mathcal{U}$ is the set of "features", $m_u(A) = \sum_{a \in A} m_u(a)$ with $m_u(a)$ measuring the degree that the article $a$ possesses the feature $u \in \mathcal{U}$. In the experiments, we define $\mathcal{U}$ as the set of all words occurred in the entire data set and $m_u(a)$ as the number of occurrences of the word $u \in \mathcal{U}$ in the article $a$. $f_{\text{fea}}$ is in the form of a sum of concave over modular functions, hence is monotone submodular [39]. The class of feature-based submodular function has been widely applied to model the utility of a data subset on a number of tasks, including speech data subset selection [47, 49], and image summarization [42]. Moreover $f_{\text{fea}}$ has been shown in [46] to model the log-likelihood of a data subset for a Naïve Bayes classifier.

We compare the submodular partitioning with the random partitioning for $m = 5$ and $m = 10$. We test with 10 instances of random partitioning. We first examine how balanced the sizes of the blocks yielded by submodular partitioning are. This is important since if the block sizes vary a lot in a partition, the computational loads across the blocks are imbalanced, and the actual efficiency of the parallel system is significantly reduced. Fortunately we observe that submodular partitioning yields very balanced partition. The sizes of all blocks in the resultant partitioning for $m = 5$ range between $2,225$ and $2,281$. In the case of $m = 10$ the maximum block is of size $1,140$, while the smallest block has $1,109$ items.

Figure 3: Distributed convex optimization tested on 20Newsgroups.

The comparison between submodular partitioning and random partitioning in terms of the accuracy of the model attained at any iteration is shown in Fig 3. For $m = 10$ we also run an instance on an adversarial partitioning, where each block is formed by grouping every two of the 20 classes in the training data. We observe submodular partitioning converges faster than the random partitioning, both of which perform significantly better than the adversarial partition. In particular significant and consistent improvement over the best of 10 random instances is achieved by the submodular partition across all iterations when $m = 5$.

**Distributed Deep Neural Network Training:** Next we evaluate our framework on distributed neural network training. We test on two tasks: 1) handwritten digit recognition on the MNIST database [5]; 2) phone classification on the TIMIT data.

The data for the handwritten digit recognition task consists of 60,000 training and 10,000 test samples. Each data sample is an image of handwritten digit. The training and test data are almost evenly divided into 10 different classes. For the phone classification task, the data consists of 1,124,823 training and 112,487 test samples. Each sample is a frame of speech. The training data is divided into 50 classes, each of which corresponds to a phoneme. The goal of this task to classify each speech sample into one of the 50 phone classes.

A 4-layer DNN model is applied to the MNIST experiments, and we train a 5-layered DNN for the TIMIT experiments. We apply the same distributed training procedure for both tasks. Given a

partitioning of the training data, we distributively solve $m$ instances of sub-problems in each iteration. We define each sub-problem on a separate block of the data. We employ the stochastic gradient descent as the solver on each instance of the sub-problem. In the first iteration we use a randomly generated model as the initial model shared among the $m$ sub-problems. Each sub-problem is solved with 10 epochs of the stochastic gradient decent training. We then average the weights in the $m$ resultant models to obtain a consensus model, which is used as the initial model for each sub-problem in the successive iteration. Note that this distributed training scheme is similar to the ones presented in [38].

The submodular partitioning for both tasks is obtained by solving the homogeneous case of Problem 1 ($\lambda = 0$) using GREEDMAX on a form of clustered facility location, as proposed and used in [46]. The function is defined as follows:

$$f_{\text{c-fac}}(A) = \sum_{y \in \mathcal{Y}} \sum_{v \in V^y} \max_{a \in A \cap V^y} s_{v,a} \tag{22}$$

where $s_{v,a}$ is the similarity measure between sample $v$ and $a$, $\mathcal{Y}$ is the set of class labels, and $V^y$ is the set of samples in $V$ with label $y \in \mathcal{Y}$. Note $\{V^y\}_{y \in \mathcal{Y}}$ forms a disjoint partitioning of the ground set $V$. In both the MNIST and TIMIT experiments we compute the similarity $s_{v,a}$ as the RBF kernel between the feature representation of $v$ and $a$. [46] show that $f_{\text{c-fac}}$ models the log-likelihood of a data subset for a Nearest Neighbor classifier. They also empirically demonstrate the efficacy of $f_{\text{c-fac}}$ in the case of neural network based classifiers.

Figure 4: MNIST                    Figure 5: TIMIT

Similar to the ADMM experiment we also observe that submodular partitioning yields very balanced partitions in all cases of this experiment. In the cases of $m = 5$ and $m = 10$ for the MNIST data set the sizes of the blocks in the resultant submodular partitioning are within the range $[11991, 12012]$ and $[5981, 6019]$, respectively. For the TIMIT data set, the block sizes range between $37, 483$ and $37510$ in the case of $m = 30$, and the range of $[28121, 28122]$ is observed for $m = 40$.

We also run 10 instances of random partitioning as a baseline. The comparison between submodular partitioning and random partitioning in terms of the accuracy of the model attained at any iteration is shown in Fig 4 and 5. The adversarial partitioning, which is formed by grouping items with the same

class, cannot even be trained in both cases. Consistent and significant improvement is again achieved by submodular partitioning for all cases.

## 5.3  Problem 2 for Unsupervised Image Segmentation

Lastly we test the efficacy of Problem 2 on the task of unsupervised image segmentation. We evaluate on the Grab-Cut data set, which consists of 30 color images. Each image has ground truth foreground/background labels. By "unsupervised", we mean that no labeled data at any time in supervised or semi-supervised training, nor any kind of interactive segmentation, was used in forming or optimizing the objective. In our experiments, the image segmentation task is solved as unsupervised clustering of the pixels, where the goal is to obtain a partitioning of the pixels such that the majority of the pixels in each block share either the same foreground or the background labels.

Let $V$ be the ground set of pixels of an image, $\pi$ be an $m$-partition of the image, and $\{y_v\}_{v \in V}$ as the pixel-wise ground truth label ($y_v = \{0, 1\}$ with 0 being background and 1 being foreground). We measure the performance of the partition $\pi$ in the following two steps: (1) for each block $i$, predict $\hat{y}_v$ for all the pixels $v \in A_i^\pi$ in the block as either 0 or 1 having larger intersection with the ground truth labels, i.e., predict $\hat{y}_v = 1, \forall v \in A_i^\pi$, if $\sum_{v \in A_i^\pi} \mathbb{1}\{y_v = 1\} \geq \sum_{v \in A_i^\pi} \mathbb{1}\{y_v = 0\}$, and predict $\hat{y}_v = 0, \forall v \in A_i^\pi$ otherwise. (2) report the performance of the partition $\pi$ as the F-measure of the predicted labels $\{\hat{y}_v\}_{v \in V}$ relative to the ground truth label $\{y_v\}_{v \in V}$.

In the experiment we first preprocess the data by downsampling each image by a factor 0.25 for testing efficiency. We represent each pixel $v$ as 5-dimensional features $x_v \in \mathbb{R}^5$, including the RGB values and pixel positions. We normalize each feature within $[0, 1]$. To obtain a segmentation of each image we solve an instance of Problem 2 ($0 < \lambda < 1$) under the homogeneous setting using GENERALGREEDMIN (Alg. 9). We use the facility location function $f_{\text{fac}}$ as the objective for Problem 2. The similarity $s_{v,a}$ between the pixels $v$ and $a$ is computed as $s_{v,a} = C - \|x_v - x_a\|_2$ with $C = \max_{v,v' \in V} \|x_v - x_v'\|_2$ being the maximum pairwise Euclidean distance. Since the facility location function $f_{\text{fac}}$ is defined on a pairwise similarity graph, which requires $O(|V|^2)$ memory complexity. It becomes computationally infeasible for medium sized images. Fortunately a facility location function that is defined on a sparse $k$-nearest neighbor similarity graph performs just as well with $k$ being very sparse [45]. In the experiment, we instantiate $f_{\text{fac}}$ by a sparse 10-nearest neighbor sparse graph, where each item $v$ is connected only to its 10 closest neighbors.

A number of unsupervised methods are tested as baselines in the experiment, including $k$-means, $k$-medoids, graph cuts [6] and spectral clustering [43]. We use the RBF kernel sparse similarity matrix as the input for spectral clustering. The sparsity of the similarity matrix is $k$ and the width parameter of the RBF kernel $\sigma$. We test with various choices of $\sigma$ and $k$ and find that the setting of $\sigma = 1$ and $k = 20$ performs the best, with which we report the results. For graph cuts, we use the MATLAB implementation [3], which has a smoothness parameter $\alpha$. We tune $\alpha = 0.3$ to achieve the best performance and report the result of graph cuts using this choice.

Figure 6:

Figure 7

The proposed image segmentation method involves a hyperparameter $\lambda$, which controls the trade-off between the worst-case objective and the average-case objective. First we examine how the

performance of our method varies with different choices of $\lambda$ in Figure 6. The performance is measured as the averaged $F$-measure of a partitioning method over all images in the data set. Interestingly we observe that the performance smoothly varies as $\lambda$ increases from 0 to 1. In particular the best performance is achieved when $\lambda$ is within the range $[0.7, 0.9]$. It suggests that using only the worst-case or the average-case objective does not suffice for the unsupervised image segmentation / clustering task, and an improved result is achieved by mixing these two extreme cases. In the subsequent experiments we show only the result of our method with $\lambda = 0.2$. Next we compare the proposed approach with baseline methods on various $m$ in Figure 7. In general, each method improves as $m$ increases. Submodular partitioning method performs the best on almost all cases of $m$. Lastly we show in Figure 8 example segmentation results on several example images as well as averaged F-measure in the case of $m = 15$. We observe that submodular partitioning, in general, leads to less noisy and more coherent segmentation in comparison to the baselines.

| F-measure on all of GrabCut | Original | | | | | | | |
|---|---|---|---|---|---|---|---|---|
| 1.0 | Ground Truth | | | | | | | |
| 0.810 | k-means | | | | | | | |
| 0.823 | k-medoids | | | | | | | |
| 0.854 | Spectral Clustering | | | | | | | |
| 0.853 | Graph Cut | | | | | | | |
| **0.870** | Submodular Partitioning | | | | | | | |

Figure 8: Unsupervised image segmentation (right: some examples).

# 6 Conclusions

In this paper, we considered two novel mixed robust/average-case submodular partitioning problems, which generalize four well-known problems: submodular fair allocation (SFA), submodular load balancing (SLB), submodular welfare problem (SWP), and submodular multiway partition (SMP). While the average case problems, i.e., SWP and SMP, admit efficient and tight algorithms, existing approaches for the worst case problems, i.e., SFA and SLB, are, in general, not scalable. We bridge this gap by providing several new algorithms that not only scale to large data sets but that also achieve comparable theoretical guarantees. Moreover we provide a number of efficient frameworks for solving the general mixed robust/average-case submodular partitioning problems. We also demonstrate that submodular partitioning is applicable in a number of machine learning problems involving distributed optimization, computational load balancing, and unsupervised image segmentation. Lastly we empirically show the effectiveness of the proposed algorithms on these machine learning tasks.

## Footnotes

[1]Similar sub-categorizations have been called the "uniform" vs. the "non-uniform" case in the past [40, 17].

[2]Results obtained in this paper are marked as ∗. Methods for only the homogeneous setting are marked as †.

[3]Results obtained in this paper are marked as *. Methods for only the homogeneous setting are marked as †.

[4]Data set is obtained at http://qwone.com/~jason/20Newsgroups/

[5]Data set is obtained at yann.lecun.com/exdb/mnist

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

## Appendix

**Proof for Theorem 2.1**

**Theorem.** *If $f_1$ and $f_2$ are monotone submodular, $\min\{f_1(A), f_2(V \setminus A)\}$ is also submodular.*

*Proof.* To prove the Theorem, we show a more general result: Let $f$ and $h$ be submodular, and $f - h$ be either monotone increasing or decreasing, then $g(S) = \min\{f(S), h(S)\}$ is also submodular. The

Theorem follows by this result, since $f(S) = f_1(S)$ and $h(S) = f_2(V \setminus S)$ are both submodular and $f(S) - h(S) = f_1(S) - f_2(V \setminus S)$ is monotone increasing.

In order to show $g$ is submodular, we prove that $g$ satisfies the following:
$$g(S) + g(T) \geq g(S \cap T) + g(S \cup T), \forall S, T \subseteq V \tag{23}$$
If $g$ agrees with either $f$ or $h$ on both $S$ and $T$, and since
$$f(S) + f(T) \geq f(S \cup T) + f(S \cap T) \tag{24}$$
$$h(S) + h(T) \geq h(S \cup T) + h(S \cap T), \tag{25}$$
$$\tag{26}$$

Eqn 23 follows.

Otherwise, w.l.o.g. we consider $g(S) = f(S)$ and $g(T) = h(T)$. For the case where $f - h$ is monotone non-decreasing, consider the following:
$$
\begin{aligned}
g(S) + g(T) &= f(S) + h(T) & \text{(27)}\\
&\geq f(S \cup T) + f(S \cap T) - (f(T) - h(T)) \text{ // submodularity of } f & \text{(28)}\\
&\geq f(S \cup T) + f(S \cap T) - (f(S \cup T) - h(S \cup T)) \text{ // monotonicity of } f - h & \text{(29)}\\
&= f(S \cap T) + h(S \cup T) & \text{(30)}\\
&\geq g(S \cap T) + g(S \cup T). & \text{(31)}
\end{aligned}
$$

Similarly, for the case where $f - h$ is monotone non-increasing, consider the following:
$$
\begin{aligned}
g(S) + g(T) &= f(S) + h(T) & \text{(32)}\\
&\geq h(S \cap T) + h(S \cup T) + (f(S) - h(S)) \text{ // submodularity of } h & \text{(33)}\\
&\geq h(S \cup T) + h(S \cap T) + (f(S \cup T) - h(S \cup T)) \text{ // monotonicity of } f - h & \text{(34)}\\
&= h(S \cap T) + f(S \cup T) & \text{(35)}\\
&\geq g(S \cap T) + g(S \cup T). & \text{(36)}
\end{aligned}
$$
$\square$

**Proof for Theorem 2.2**

**Theorem.** *Under the homogeneous setting ($f_i = f$ for all $i$), GREEDMAX is guaranteed to find a partition $\hat{\pi}$ such that*
$$\min_{i=1,\ldots,m} f(A_i^{\hat{\pi}}) \geq \frac{1}{m} \max_{\pi \in \Pi} \min_{i=1,\ldots,m} f(A_i^{\pi}). \tag{37}$$

*Proof.* We prove that the guarantee of $1/m$, in fact, even holds for a streaming version of the greedy algorithm (STREAMGREED, see Alg. 10). In particular, we show that STREAMGREED provides a factor of $1/m$ for SFA under the homogeneous setting. Theorem 2.2 then follows since GREEDMAX can be seen as running STREAMGREED with a specific order.

---

**Algorithm 10:** STREAMGREED

---

Input: $V = \{v_1, v_2, \ldots, v_n\}, f, m$
Initialize: $A_1 =, \ldots, = A_m = \emptyset, k = 1$
**while** $k \leq n$ **do**
$\quad i^* \in \text{argmin}_j f(A_j)$
$\quad A_{i^*} \leftarrow A_{i^*} \cup \{v_k\}$
$\quad k \leftarrow k + 1$

---

To prove the guarantee for STREAMGREED, we consider the resulting partitioning after an instance of STREAMGREED: $\pi = (A_1^{\pi} \cup A_2^{\pi}, \ldots, A_m^{\pi})$. For simplicity of notation, we write $A_i^{\pi}$ as $A_i$ for each $i$ in the remaining proof. We refer $OPT$ to the optimal solution, i.e., $OPT = \max_{\pi} \min_i f(A_i)$. W.l.o.g., we assume $f(A_1) = \min_i f(A_i)$. Let $a_i$ be the last item to be chosen in block $A_i$ for $i = 2, \ldots, m$.

Claim 1:
$$OPT \leq f(V \setminus \{a_2, \ldots, a_m\}) \tag{38}$$

To show this claim, consider the following: If we enlarge the singleton value of $a_i, i = 2, \ldots, m$, we obtain a new submodular function:

$$f'(A) = f(A) + \alpha \sum_{i=2}^{m} |A \cap a_i|, \tag{39}$$

where $\alpha$ is sufficiently large. Then running STREAMGREED on $f'$ with the same ordering of the incoming items leads to the same solution, since only the gain of the last added item for each block is changed.

Note that $f'(A) \geq f(A), \forall A \subseteq V$, we then have $\max_\pi \min_i f'(A_i^\pi) \geq OPT$. The optimal partitioning for $f'$ can be easily obtained as $\pi' = (V \setminus \{a_2, \ldots, a_m\}, a_2, \ldots, a_m)$. Therefore, we have that

$$OPT \leq \max_\pi \min_i f'(A_i^\pi) = f'(V \setminus \{a_2, \ldots, a_m\}) = f(V \setminus \{a_2, \ldots, a_m\}). \tag{40}$$

Lastly, we have that $f(A_1) \geq f(A_i \setminus a_i)$ for any $i = 2, \ldots, m$ due to the procedure of STREAM-GREED. Therefore we have the following:

$$f(A_1) \geq \frac{1}{m}(f(A_1) + \sum_{i=2}^{m} f(A_i \setminus a_i)) \tag{41}$$

$$\geq \frac{1}{m} f(V \setminus \{a_2, \ldots, a_m\}) \text{ // submodularity of } f \tag{42}$$

$$\geq \frac{1}{m} OPT \text{ // Claim 1} \tag{43}$$

$\square$

**Proof for Theorem 2.3**

**Theorem.** *Given $\epsilon$, $\alpha$ and any $0 < \delta < \alpha$, GREEDSAT finds a partition such that at least $\lceil m(\alpha - \delta) \rceil$ blocks receive utility at least $\frac{\delta}{1-\alpha+\delta}(\max_\pi \min_i f_i(A_i^\pi) - \epsilon)$.*

*Proof.* When GREEDSAT terminates, it identifies a $c_{\min}$ such that the returned solution $\hat{\pi}^{c_{\min}}$ satisfies $\bar{F}^{c_{\min}}(\hat{\pi}^{c_{\min}}) \geq \alpha c_{\min}$. Also it identifies a $c_{\max}$ such that the returned solution $\hat{\pi}^{c_{\max}}$ satisfies $\bar{F}^{c_{\max}}(\hat{\pi}^{c_{\max}}) < \alpha c_{\max}$. The gap between $c_{\max}$ and $c_{\min}$ is bounded by $\epsilon$, i.e., $c_{\max} - c_{\min} \leq \epsilon$.

Next, we prove that there does not exist any partitioning $\pi$ that satisfies $\min_i f_i(A_i^\pi) \geq c_{\max}$, i.e., $c_{\max} \geq \max_{\pi \in \Pi} \min_i f_i(A_i^\pi)$.

Suppose otherwise, i.e., $\exists \pi^* : \min_i f_i(A_i^{\pi^*}) = c_{\max} + \gamma$ with $\gamma \geq 0$. Let $c = c_{\max} + \gamma$, consider the intermediate objective $\bar{F}^c(\pi) = \frac{1}{m} \sum_{i=1}^{m} \min\{f_i(A_i^\pi), c\}$, we have that $\bar{F}^c(\pi^*) = c$. An instance of the algorithm for SWP on $\bar{F}^c$ is guaranteed to lead to a solution $\hat{\pi}^c$ such that $\bar{F}^c(\hat{\pi}^c) \geq \alpha c$. Since $c \geq c_{\max}$, it should follow that the returned solution $\hat{\pi}^{c_{\max}}$ for the value $c_{\max}$ also satisfies $\bar{F}^{c_{\max}}(\hat{\pi}) \geq \alpha c_{\max}$. However it contradicts with the termination criterion of GREEDSAT. Therefore, we prove that $c_{\max} \geq \max_{\pi \in \Pi} \min_i f_i(A_i^\pi)$, which indicates that $c_{\min} \geq c_{\max} - \epsilon \geq \max_{\pi \in \Pi} \min_i f_i(A_i^\pi) - \epsilon$.

Let $c^* = \frac{c_{\max} + c_{\min}}{2}$ and the partitioning returned by running for $c^*$ be $\hat{\pi}$ (the final output partitioning from GREEDSAT). We have that $\bar{F}^{c^*}(\hat{\pi}) \geq \alpha c^*$, we are going to show that for any $0 < \delta < \alpha$, at least a $\lceil m(\alpha - \delta) \rceil$ blocks given by $\hat{\pi}$ receive utility larger or equal to $\frac{\delta}{1-\alpha+\delta} c^*$.

Just to restate the problem: we say that the $i^{\text{th}}$ block is $(\alpha, \delta)$-good if $f_i(A_i^{\hat{\pi}}) \geq \frac{\delta}{1-\alpha+\delta} c^*$. Then the statement becomes: Given $0 < \delta < \alpha$, there is at least $m \lceil \alpha - \delta \rceil$ blocks that are $(\alpha, \delta)$-good.

To prove this statement, we assume, by contradiction, that there is strictly less than $\lceil m(\alpha - \delta) \rceil$ $(\alpha, \delta)$-good blocks. Denote the number of $(\alpha, \delta)$-good blocks as $m_{\text{good}}$. Then we have that $m_{\text{good}} \leq \lceil m(\alpha - \delta) \rceil - 1 < m(\alpha - \delta)$. Let $\theta = \frac{m_{\text{good}}}{m}$ be the fraction of $(\alpha, \delta)$ good blocks, then we have that $0 \leq \theta < (\alpha - \delta) < 1$. The remaining fraction $(1 - \theta)$ of blocks are not good, i.e., they have

valuation strictly less than $\frac{\delta}{1-\alpha+\delta}c^*$. Then, consider the following:

$$\bar{F}^{c^*}(\hat{\pi}) = \frac{1}{m}\sum_{i=1}^{m}\min\{f_i(A_i^{\hat{\pi}}), c^*\} \tag{44}$$

$$\overset{(a)}{<} \theta c^* + (1-\theta)\frac{\delta}{1-\alpha+\delta}c^* \tag{45}$$

$$= \frac{\delta}{1-\alpha+\delta}c^* + \frac{1-\alpha}{1-\alpha+\delta}\theta c^* \tag{46}$$

$$\overset{(b)}{<} \frac{\delta}{1-\alpha+\delta}c^* + \frac{1-\alpha}{1-\alpha+\delta}(\alpha-\delta)c^* \tag{47}$$

$$= \alpha c^* \tag{48}$$

Inequality (a) follows since good blocks are upper bounded by $c^*$, and not good blocks have values upper bounded by $\frac{\delta}{1-\alpha+\delta}c^*$. Inequality (b) follows by the assumption on $\theta$. This therefore contradicts the assumption that $\bar{F}^{c^*}(\hat{\pi}) \geq \alpha c^*$, hence the statement is true.

This statement can also be proved using a different strategy. Let $f_i^{c^*} = \min\{f_i(A_i^{\hat{\pi}}), c^*\}$ and $f_i = f_i(A_i^{\hat{\pi}})$ for all $i$. For any $0 \leq \beta \leq 1$ the following holds:

$$\alpha c^* \leq \bar{F}^{c^*}(\hat{\pi}) = \frac{1}{m}\sum_i f_i^{c^*} \leq \frac{1}{m}f_i = \frac{1}{m}\sum_{i:f_i<\beta c^*} f_i + \frac{1}{m}\sum_{i:f_i\geq\beta c^*} f_i < \frac{1}{m}m_{\text{bad}}\beta c^* + \frac{1}{m}m_{\text{good}}c^*$$
$$\tag{49}$$

where $m = m_{\text{bad}} + m_{\text{good}}$ and $m_{\text{good}}$ are the number that are $\beta$-good (i.e., $i$ is $\beta$-good if $f_i \geq \beta c^*$). The goal is to place a lower bound on $m_{\text{good}}$. From the above

$$\alpha < (1 - \frac{m_{\text{good}}}{m})\beta + \frac{m_{\text{good}}}{m} \tag{50}$$

which means

$$m_{\text{good}} \geq \lceil \frac{\alpha-\beta}{1-\beta}m \rceil \tag{51}$$

Let $\beta = \frac{\delta}{1-\alpha+\delta}$, the statement immediately follows.

Note $c^* = \frac{c_{\min}+c_{\max}}{2} \geq c_{\max} - \epsilon \geq \max_{\pi\in\Pi}\min_i f_i(A_i^{\pi}) - \epsilon$. Combining pieces together, we have shown that at least $\lceil m(\alpha - \delta) \rceil$ blocks given by $\hat{\pi}$ receive utility larger or equal to $\frac{\delta}{1-\alpha+\delta}(\max_{\pi\in\Pi}\min_i f_i(A_i^{\pi}) - \epsilon)$. $\square$

**Proof for Theorem 2.4**

**Theorem.** MMAX *achieves a worst-case guarantee of* $O(\min_i \frac{1+(|A_i^{\hat{\pi}}|-1)(1-\kappa_{f_i}(A_i^{\hat{\pi}}))}{|A_i^{\hat{\pi}}|\sqrt{m}\log^3 m})$, *where* $\hat{\pi} = (A_1^{\hat{\pi}}, \cdots, A_m^{\hat{\pi}})$ *is the partition obtained by the algorithm, and* $\kappa_f(A) = 1 - \min_{v\in V}\frac{f(v|A\backslash v)}{f(v)} \in [0,1]$ *is the curvature of a submodular function* $f$ *at* $A \subseteq V$.

*Proof.* We assume the approximation factor of the algorithm for solving the modular version of Problem 1 is $\alpha = O(\frac{1}{\sqrt{m}\log^3 m})$ [2]. For notation simplicity, we write $\hat{\pi} = (\hat{A}_1, \ldots, \hat{A}_m)$ as the resulting partition after the first iteration of MMAX, and $\pi^* = (A_1^*, \ldots, A_m^*)$ as its optimal solution. Note that first iteration suffices to yield the performance guarantee, and the subsequent iterations are designed so as to improve the empirical performance. Since the proxy function for each function $f_i$ used for the first iteration are the simple modular upper bound with the form: $h_i(X) = \sum_{j\in X} f_i(j)$.

Given the curvature of each submodular function $f_i$, we can tightly bound a submodular function $f_i$ in the following form [24]:

$$f_i(X) \leq h_i(X) \leq \frac{|X|}{1+(|X|-1)(1-\kappa_{f_i}(X))}f_i(X), \forall X \subseteq V \tag{52}$$

Consider the following:

$$\min_i f_i(\hat{A}_i) \geq \min_i \frac{1}{\frac{|\hat{A}_i|}{1+(|\hat{A}_i|-1)(1-\kappa_{f_i}(\hat{A}_i))}} h_i(\hat{A}_i) \tag{53}$$

$$\geq \min_i \frac{1}{\frac{|\hat{A}_i|}{1+(|\hat{A}_i|-1)(1-\kappa_{f_i}(\hat{A}_i))}} \min_i h_i(\hat{A}_i) \tag{54}$$

$$\geq \alpha \min_i \frac{1+(|\hat{A}_i|-1)(1-\kappa_{f_i}(\hat{A}_i))}{|\hat{A}_i|} \min_i h_i(A_i^*) \tag{55}$$

$$\geq \alpha \min_i \frac{1+(|\hat{A}_i|-1)(1-\kappa_{f_i}(\hat{A}_i))}{|\hat{A}_i|} \min_i f_i(A_i^*) \tag{56}$$

$$= O(\min_i \frac{1+(|\hat{A}_i|-1)(1-\kappa_{f_i}(\hat{A}_i))}{|\hat{A}_i|\sqrt{m}\log^3 m}) \min_i f_i(A_i^*). \tag{57}$$

$\square$

### Proof for Theorem 2.5

**Theorem.** *Suppose there exists an algorithm for solving the modular version of SFA with an approximation factor $\alpha \leq 1$, we have that*

$$\min_i f_i(A_i^{\pi_t}) \geq \alpha \min_i f_i(A_i^{\pi_{t-1}}). \tag{58}$$

*Proof.* Consider the following:

$$\min_i f_i(A_i^{\pi_{t-1}}) = \min_i h_i(A_i^{\pi_{t-1}}) // \text{ tightness of modular lower bound.} \tag{59}$$

$$\leq \alpha \min_i h_i(A_i^{\pi_t}) // \text{ approximation factor of the modular SFA.} \tag{60}$$

$$\leq \alpha h_j(A_j^{\pi_t}) // j \in \operatorname*{argmin}_i f_i(A_i^{\pi_t}) \tag{61}$$

$$\leq \alpha f_j(A_j^{\pi_t}) // h_j(A_j^{\pi_{t-1}}) \text{ upper bounds } f_j \text{ everywhere.} \tag{62}$$

$$= \alpha \min_i f_i(A_i^{\pi_t}) \tag{63}$$

$\square$

### Proof for Theorem 2.6

**Theorem.** *For any $\epsilon > 0$, SLB cannot be approximated to a factor of $(1-\epsilon)m$ for any $m = o(\sqrt{n/\log n})$ with polynomial number of queries even under the homogeneous setting.*

*Proof.* We use the same proof techniques as in [40]. Consider two submodular functions:

$$f_1(S) = \min\{|S|, \alpha\}; \tag{64}$$

$$f_2(S) = \min\{\sum_{i=1}^m \min\{\beta, |S \cap V_i|\}, \alpha\}; \tag{65}$$

where $\{V_i\}_{i=1}^m$ is a uniformly random partitioning of $V$ into $m$ blocks, $\alpha = \frac{n}{m}$ and $\beta = \frac{n}{m^2(1-\epsilon)}$. To be more precise about the uniformly random partitioning, we assign each item into any one of the $m$ blocks with probability $1/m$. It can be easily verified that $OPT_1 = \min_{\pi \in \Pi} \max_i f_1(A_i^\pi) = n/m$ and $OPT_2 = \min_{\pi \in \Pi} \max_i f_2(A_i^\pi) = \frac{n}{m^2(1-\epsilon)}$. The gap is then $\frac{OPT_1}{OPT_2} = m(1-\epsilon)$.

Next, we show that $f_1$ and $f_2$ cannot be distinguished with $n^{\omega(1)}$ number of queries.

Since $f_1(S) \geq f_2(S)$ holds for any $S$, this is equivalent as showing $P\{f_1(S) > f_2(S)\} < n^{-\omega(1)}$.

As shown in [40], $P\{f_1(S) > f_2(S)\}$ is maximized when $|S| = \alpha$. It suffices to consider only the case of $|S| = \alpha$ as follows:

$$P\{f_1(S) > f_2(S) : |S| = \alpha\} = P\{\sum_{i=1}^{m} \min\{\beta, |S \cap V_i|\} < \alpha : |S| = \alpha\} \tag{66}$$

The necessary condition for $\sum_{i=1}^{m} \min\{\beta, |S \cap V_i|\} < \alpha$ is that $|S \cap V_i| > \beta$ is satisfied for some $i$. Using the Chernoff bound, we have that for any $i$, it holds $P\{|S \cap V_i| > \beta\} \le e^{-\frac{\epsilon^2 n}{3m^2}} = n^{-\omega(1)}$ when $m = o(\sqrt{n/\log n})$. Using the union bound, it holds that the probability for any one block $V_i$ such that $|S \cap V_i| > \beta$ is also upper bounded by $n^{-\omega(1)}$. Combining all pieces together, we have the following:

$$P\{f_1(S) > f_2(S)\} \le n^{-\omega(1)}. \tag{67}$$

Finally, we come to prove the Theorem. Suppose the goal is to solve an instance of SLB with $f_2$. Since $f_1$ and $f_2$ are hard to distinguish with polynomial number of function queries, any polynomial time algorithm for solving $f_2$ is equivalent to solving for $f_1$. However, the optimal solution for $f_1$ is $\alpha = \frac{n}{m}$, whereas the optimal solution for $f_2$ is $\beta = \frac{n}{m^2(1-\epsilon)}$. Therefore, no polynomial time algorithm can find a solution with a factor $m(1 - \epsilon)$ for SLB in this case. $\qquad\square$

### Proof for Theorem 2.7

**Theorem.** LOVÁSZROUND *is guaranteed to find a partition* $\hat{\pi} \in \Pi$ *such that* $\max_i f_i(A_i^{\hat{\pi}}) \le m \min_{\pi \in \Pi} \max_i f_i(A_i^{\pi})$.

*Proof.* It suffices to bound the performance loss at the step of rounding the fractional solution $\{x_i^*\}_{i=1}^{m}$, or equivalently, the following:

$$\max_i \tilde{f}_i(x_i^*) \ge \frac{1}{m} \max_i f_i(A_i), \tag{68}$$

where $\{A_i\}_{i=1}^{m}$ is the resulting partitioning after the rounding. To show Eqn 68, it suffices to show that $\tilde{f}_i(x_i^*) \ge \frac{1}{m} f_i(A_i)$ for all $i = 1, \dots, m$. Next, consider the following:

$$f_i(A_i) = \tilde{f}_i(1_{A_i}) = m\tilde{f}_i(\frac{1}{m}1_{A_i})// \text{ positive homogeneity of Lovász extension} \tag{69}$$

For any item $v_j \in A_i$, we have $x_i^*(j) \ge \frac{1}{m}$, since $\sum_{i=1}^{m} x_i^*(j) \ge 1$ and $x_i^*(j) = \max_{i'} x_{i'}(j)$. Therefore, we have $\frac{1}{m}1_{A_i} \le x_i^*$. Since $f_i$ is monotone, its extension $\tilde{f}_i$ is also monotone. As a result, $f_i(A_i) = m\tilde{f}_i(\frac{1}{m}1_{A_i}) \le m\tilde{f}_i(x_i^*)$.

$\qquad\square$

### Proof for Theorem 2.8

**Theorem.** MMIN *achieves a worst-case guarantee of* $(2 \max_i \frac{|A_i^{\pi^*}|}{1 + (|A_i^{\pi^*}| - 1)(1 - \kappa_{f_i}(A_i^{\pi^*}))})$, *where* $\pi^* = (A_1^{\pi^*}, \cdots, A_m^{\pi^*})$ *denotes the optimal partition.*

*Proof.* Let $\alpha = 2$ be the approximation factor of the algorithm for solving the modular version of Problem 2 [32]. For notation simplicity, we write $\hat{\pi} = (\hat{A}_1, \dots, \hat{A}_m)$ as the resulting partition after the first iteration of MMIN, and $\pi^* = (A_1^*, \dots, A_m^*)$ as its optimal solution. Again the first iteration suffices to yield the performance guarantee, and the subsequent iterations are designed so as to improve the empirical performance. Since the supergradients for each function $f_i$ used for the first iteration are the simple modular upper bound with the form: $h_i(X) = \sum_{j \in X} f_i(j)$, we can again tightly bound a submodular function $f_i$ in the following form:

$$f_i(X) \le h_i(X) \le \frac{|X|}{1 + (|X| - 1)(1 - \kappa_{f_i}(X))} f_i(X), \forall X \subseteq V \tag{70}$$

Consider the following:
$$\max_i f_i(\hat{A}_i) \leq \max_i h_i(\hat{A}_i) \tag{71}$$
$$\leq \alpha \max_i h_i(A_i^*) \tag{72}$$
$$\leq \alpha \max_i \frac{|A_i^*|}{1 + (|A_i^*| - 1)(1 - \kappa_{f_i}(A_i^*))} f_i(A_i^*) \tag{73}$$
$$\leq \alpha \max_i \frac{|A_i^*|}{1 + (|A_i^*| - 1)(1 - \kappa_{f_i}(A_i^*))} \max_i f_i(A_i^*) \tag{74}$$
$$\square$$

## Proof for Theorem 2.9

**Theorem.** *Suppose there exists an algorithm for solving the modular version of SLB with an approximation factor $\alpha \geq 1$, we have for each iteration $t$ that*
$$\max_i f_i(A_i^{\pi_t}) \leq \alpha \max_i f_i(A_i^{\pi_{t-1}}). \tag{75}$$

*Proof.* The proof is symmetric to the one for Theorem 2.5. $\qquad\square$

## Proof for Theorem 3.1

We prove separately for Problem 1 and Problem 2.

**Theorem.** *Given an instance of Problem 1 with $0 < \lambda < 1$, COMBSFASWP provides an approximation guarantee of $\max\{\min\{\alpha, \frac{1}{m}\}, \frac{\beta\alpha}{\bar{\lambda}\beta+\alpha}, \lambda\beta\}$ in the homogeneous case, and a factor of $\max\{\frac{\beta\alpha}{\bar{\lambda}\beta+\alpha}, \lambda\beta\}$ in the heterogeneous case, where $\alpha$ and $\beta$ are the approximation factors of ALGWC and ALGAC for SFA and SWP respectively.*

*Proof.* We first prove the result for heterogeneous setting. For notation simplicity we write the worst-case objective as $F_1(\pi) = \min_{i=1,\dots,m} f(A_i^\pi)$ and the average-case objective as $F_2(\pi) = \frac{1}{m} \sum_{i=1,\dots,m} f(A_i^\pi)$.

Suppose ALGWC outputs a partition $\hat{\pi}_1$ and ALGAC outputs a partition $\hat{\pi}_2$. Let $\pi^* \in \arg\max_{\pi \in \Pi} \bar{\lambda} F_1(\pi) + \lambda F_2(\pi)$ be the optimal partition.

We use the following facts:

Fact1
$$F_1(\hat{\pi}_1) \geq \alpha F_1(\pi) \tag{76}$$
Fact2
$$F_2(\pi_2) \geq \beta F_2(\pi) \tag{77}$$
Fact3
$$F_1(\pi) \leq F_2(\pi) \tag{78}$$

Then we have that

$$\bar{\lambda} F_1(\hat{\pi}_2) + \lambda F_2(\hat{\pi}_2) \geq \lambda F_2(\hat{\pi}_2) \tag{79}$$
$$\geq \lambda\beta F_2(\pi^*) \tag{80}$$
$$\geq \lambda\beta \left[ \bar{\lambda} F_1(\pi^*) + \lambda F_2(\pi^*) \right] \tag{81}$$

and

$$\bar{\lambda}F_1(\hat{\pi}_1) + \lambda F_2(\hat{\pi}_1) \geq \mu\left[\bar{\lambda}F_1(\hat{\pi}_1) + \lambda F_2(\hat{\pi}_1)\right] + (1-\mu)\left[\bar{\lambda}F_1(\hat{\pi}_1) + \lambda F_2(\hat{\pi}_1)\right] \tag{82}$$

$$\geq \mu\left[\bar{\lambda}\alpha F_1(\pi^*) + \lambda\alpha F_1(\pi^*)\right] + (1-\mu)\left[0 + \lambda\beta F_2(\pi^*)\right] \tag{83}$$

$$\geq \frac{\mu\alpha}{\bar{\lambda}}\bar{\lambda}F_1(\pi^*) + (1-\mu)\beta\lambda F_2(\pi^*) \tag{84}$$

$$\geq \min\{\frac{\mu\alpha}{\bar{\lambda}}, (1-\mu)\beta\}\left[\bar{\lambda}F_1(\pi^*) + \lambda F_2(\pi^*)\right] \tag{85}$$

$\min\{\frac{\mu\alpha}{\bar{\lambda}}, (1-\mu)\beta\}$ is a function over $0 \leq \mu \leq 1$ and $\mu^* \in \arg\max_\mu \min\{\frac{\mu\alpha}{\bar{\lambda}}, (1-\mu)\beta\}$. It is easy to show

$$\mu^* = \frac{\bar{\lambda}\beta}{\bar{\lambda}\beta + \alpha} \tag{86}$$

$$\max_\mu \min\{\frac{\mu\alpha}{\bar{\lambda}}, (1-\mu)\beta\} = \frac{\beta\alpha}{\bar{\lambda}\beta + \alpha} \tag{87}$$

$$\bar{\lambda}F_1(\hat{\pi}_1) + \lambda F_2(\hat{\pi}_1) \geq \frac{\beta\alpha}{\bar{\lambda}\beta + \alpha}\left[\bar{\lambda}F_1(\pi^*) + \lambda F_2(\pi^*)\right] \tag{88}$$

Taking the max over the two bounds leads to

$$\max\{\bar{\lambda}F_1(\hat{\pi}_1) + \lambda F_2(\hat{\pi}_1), \bar{\lambda}F_1(\hat{\pi}_2) + \lambda F_2(\hat{\pi}_2)\} \geq \max\{\frac{\beta\alpha}{\bar{\lambda}\beta + \alpha}, \lambda\beta\}\max_{\pi\in\Pi}\bar{\lambda}F_1(\pi) + \lambda F_2(\pi) \tag{89}$$

Next we are going to show the result for the homogeneous setting. We have the following facts that hold for arbitrary partition $\pi$:

$$F_1(\hat{\pi}_1) \geq \alpha F_1(\pi), \quad F_2(\hat{\pi}_2) \geq \beta F_2(\pi) \tag{90}$$

$$F_1(\pi) \leq F_2(\pi), \quad F_2(\pi_1) \geq \frac{1}{m}F_2(\pi) \tag{91}$$

Consider the following:

$$\bar{\lambda}F_1(\hat{\pi}_1) + \lambda F_2(\hat{\pi}_1) \geq \alpha\bar{\lambda}F_1(\pi^*) + \frac{\lambda}{m}F_2(\pi^*) \tag{92}$$

$$\geq \min\{\alpha, \frac{1}{m}\}\left[\bar{\lambda}F_1(\pi^*) + \lambda F_2(\pi^*)\right] \tag{93}$$

and

$$\bar{\lambda}F_1(\hat{\pi}_2) + \lambda F_2(\hat{\pi}_2) \geq \lambda F_2(\hat{\pi}_2) \tag{94}$$

$$\geq \lambda\beta F_2(\pi^*) \tag{95}$$

$$\geq \lambda\beta\left[\bar{\lambda}F_1(\pi^*) + \lambda F_2(\pi^*)\right] \tag{96}$$

Taking the max over the two bounds and the result shown in Eqn 88 gives the following:

$$\max\{\bar{\lambda}F_1(\hat{\pi}_1) + \lambda F_2(\hat{\pi}_1), \bar{\lambda}F_1(\hat{\pi}_2) + \lambda F_2(\hat{\pi}_2)\} \geq \max\{\min\{\alpha, \frac{1}{m}\}, \frac{\beta\alpha}{\bar{\lambda}\beta + \alpha}, \lambda\beta\}\max_{\pi\in\Pi}\bar{\lambda}F_1(\pi) + \lambda F_2(\pi). \tag{97}$$

$\square$

**Theorem.** COMBSLBSMP *provides an approximation guarantee of* $\min\{m, \frac{m\alpha}{m\bar{\lambda}+\lambda}, \beta(m\bar{\lambda}+\lambda)\}$ *in the* homogeneous *case, and a factor of* $\min\{\frac{m\alpha}{m\bar{\lambda}+\lambda}, \beta(m\bar{\lambda}+\lambda)\}$ *in the* heterogeneous *case, for Problem 2 with* $0 \leq \lambda \leq 1$.

*Proof.* Let $\hat{\pi}_1$ be the solution of ALGWC and $\hat{\pi}_2$ be the solution of ALGAC. Let $\pi^* \in \arg\min_{\pi\in\Pi}\bar{\lambda}F_1(\pi) + \lambda F_2(\pi)$ be the optimal partition. The following facts hold for all $\pi \in \Pi$:

Fact1

$$F_1(\hat{\pi}_1) \leq \alpha F_1(\pi) \tag{98}$$

Fact2

$$F_2(\hat{\pi}_2) \leq \beta F_2(\pi) \tag{99}$$

Fact3

$$F_2(\pi) \leq F_1(\pi) \leq mF_2(\pi) \tag{100}$$

Then we have the following:

$$\bar{\lambda}F_1(\hat{\pi}_1) + \lambda F_2(\hat{\pi}_1) \leq F_1(\hat{\pi}_1) \tag{101}$$

$$\leq \alpha F_1(\pi^*) \tag{102}$$

$$\leq \frac{\alpha}{\bar{\lambda} + \frac{\lambda}{m}} \left[ \bar{\lambda}F_1(\pi^*) + \frac{\lambda}{m}F_1(\pi^*) \right] \tag{103}$$

$$\leq \frac{m\alpha}{m\bar{\lambda} + \lambda} \left[ \bar{\lambda}F_1(\pi^*) + \lambda F_2(\pi^*) \right] \tag{104}$$

and

$$\bar{\lambda}F_1(\hat{\pi}_2) + \lambda F_2(\hat{\pi}_2) \leq \beta m \bar{\lambda} F_2(\pi^*) + \beta \lambda F_2(\pi^*) \tag{105}$$

$$\leq (m\bar{\lambda} + \lambda)\beta F_2(\pi^*) \tag{106}$$

$$\leq (m\bar{\lambda} + \lambda)\beta \left[ \bar{\lambda}F_1(\pi^*) + \lambda F_2(\pi^*) \right] \tag{107}$$

$$\tag{108}$$

Taking the minimum over the two leads to the following:

$$\min\{\bar{\lambda}F_1(\hat{\pi}_1) + \lambda F_2(\hat{\pi}_1), \bar{\lambda}F_1(\hat{\pi}_2) + \lambda F_2(\hat{\pi}_2)\} \leq \min\{\frac{m\alpha}{m\bar{\lambda} + \lambda}, \beta(m\bar{\lambda} + \lambda)\} \min_{\pi \in \Pi} \bar{\lambda}F_1(\pi) + \lambda F_2(\pi) \tag{109}$$

Equation 109 gives us a bound for both the homogeneous setting and the heterogeneous settings.

Furthermore, in the homogeneous setting, for arbitrary partition $\pi$, we have

$$\bar{\lambda}F_1(\pi) + \lambda F_2(\pi) \leq m \min_{\pi \in \Pi} \bar{\lambda}F_1(\pi) + \lambda F_2(\pi) \tag{110}$$

and we can tighten the bound for the homogeneous setting as follows:

$$\min\{\bar{\lambda}F_1(\hat{\pi}_1) + \lambda F_2(\hat{\pi}_1), \bar{\lambda}F_1(\hat{\pi}_2) + \lambda F_2(\hat{\pi}_2)\} \leq \min\{m, \frac{m\alpha}{m\bar{\lambda} + \lambda}, \beta(m\bar{\lambda} + \lambda)\} \max_{\pi \in \Pi} \bar{\lambda}F_1(\pi) + \lambda F_2(\pi) \tag{111}$$

$$\square$$

**Proof for Theorem 3.2**

**Theorem.** *Given $\epsilon$, $\alpha$, and, $0 \leq \lambda \leq 1$, GENERALGREEDSAT finds a partition $\hat{\pi}$ that satisfies the following:*

$$\bar{\lambda} \min_i f_i(A_i^{\hat{\pi}}) + \lambda \frac{1}{m} \sum_{i=1}^m f_i(A_i^{\hat{\pi}}) \geq \lambda\alpha(OPT - \epsilon), \tag{112}$$

*where $OPT = \max_{\pi \in \Pi} \bar{\lambda} \min_i f_i(A_i^\pi) + \lambda \frac{1}{m} \sum_{i=1}^m f_i(A_i^\pi)$.*

*Moreover, let $F_{\lambda,i}(\pi) = \bar{\lambda}f_i(A_i^\pi) + \lambda \frac{1}{m} \sum_{j=1}^m f_j(A_j^\pi)$. Given any $0 < \delta < \alpha$, there is a set $I \subseteq \{1, \ldots, m\}$ such that $|I| \geq \lceil m(\alpha - \delta) \rceil$ and*

$$F_{i,\lambda}(\hat{\pi}) \geq \max\{\frac{\delta}{1 - \alpha + \delta}, \lambda\alpha\}(OPT - \epsilon), \forall i \in I. \tag{113}$$

*Proof.* Denote intermediate objective $\bar{F}^c(\pi) = \frac{1}{m} \sum_{i=1}^m \min\{\bar{\lambda}f_i(A_i^\pi) + \lambda \frac{1}{m} \sum_{j=1}^m f_j(A_j^\pi), c\}$. Also we define the overall objective as $F(\pi) = \bar{\lambda} \min_i f_i(A_i^\pi) + \lambda \frac{1}{m} \sum_{i=1}^m f_i(A_i^\pi)$. When the algorithm terminates, it identifies a $c_{\min}$ such that the returned solution $\hat{\pi}^{c_{\min}}$ satisfies $\bar{F}^{c_{\min}}(\hat{\pi}^{c_{\min}}) \geq \alpha c_{\min}$. Also it identifies a $c_{\max}$ such that the returned solution $\hat{\pi}^{c_{\max}}$ satisfies $\bar{F}^{c_{\max}}(\hat{\pi}^{c_{\max}}) < \alpha c_{\max}$. The gap between $c_{\max}$ and $c_{\min}$ is bounded by $\epsilon$, i.e., $c_{\max} - c_{\min} \leq \epsilon$.

Next, we prove that there does not exist any partitioning $\pi$ that satisfies $F(\pi) \geq c_{\max}$, i.e., $c_{\max} \geq OPT$.

Suppose otherwise, i.e., $\exists \pi^* : F(\pi^*) = c_{\max} + \gamma$ with $\gamma \geq 0$. Let $c = c_{\max} + \gamma$, consider the intermediate objective $\bar{F}^c(\pi)$, we have that $\bar{F}^c(\pi^*) = c$. An instance of the algorithm for SWP on $\bar{F}^c$ is guaranteed to lead to a solution $\hat{\pi}^c$ such that $\bar{F}^c(\hat{\pi}^c) \geq \alpha c$. Since $c \geq c_{\max}$, it should follow

that the returned solution $\hat{\pi}^{c_{\max}}$ for the value $c_{\max}$ also satisfies $\bar{F}^{c_{\max}}(\hat{\pi}) \geq \alpha c_{\max}$. However it contradicts with the termination criterion of GREEDSAT. Therefore, we prove that $c_{\max} \geq OPT$, which indicates that $c_{\min} \geq c_{\max} - \epsilon \geq OPT - \epsilon$.

Let $c^* = \frac{c_{\max} + c_{\min}}{2}$ and the partitioning returned by running for $c^*$ be $\hat{\pi}$ (the final output partitioning from the algorithm). We have that $\bar{F}^{c^*}(\hat{\pi}) \geq \alpha c^*$.

Next we are ready to prove the Theorem: $F(\hat{\pi}) \geq \lambda \alpha$. For simplicity of notation, we rewrite $y_i = \bar{\lambda} f_i(A_i^{\hat{\pi}}) + \lambda \frac{1}{m} \sum_{j=1}^{m} f_j(A_j^{\hat{\pi}})$ and $x_i = \min\{\bar{\lambda} f_i(A_i^{\hat{\pi}}) + \lambda \frac{1}{m} \sum_{j=1}^{m} f_j(A_j^{\hat{\pi}}), c^*\} = \min\{y_i, c^*\}$ for each $i$. Furthermore, we denote the sample mean as $\bar{x} = \frac{1}{m} \sum_{i=1}^{m} x_i$ and $\bar{y} = \frac{1}{m} \sum_{i=1}^{m} y_i$. Then, we have $F(\hat{\pi}) = \min_i y_i$ and $\bar{F}^{c^*}(\hat{\pi}) = \bar{x}$. We list the following facts to facilitate the analysis:

1. $0 \leq x_i \leq c^*$ holds for all $i$;

2. $y_i \geq \lambda \bar{y}$ holds for all $i$;

3. $x_i \geq \lambda \bar{x}$ holds for all $i$;

4. $\bar{x} \geq \alpha c^*$;

5. $x_i = \min\{y_i, c^*\}, \forall i$.

The second fact follows since

$$\bar{y} = \frac{1}{m} \sum_{i=1}^{m} y_i \tag{114}$$

$$= \frac{1}{m} \sum_{i=1}^{m} \{\bar{\lambda} f_i(A_i^{\hat{\pi}}) + \lambda \frac{1}{m} \sum_{j=1}^{m} f_j(A_j^{\hat{\pi}})\} \tag{115}$$

$$= \frac{1}{m} \sum_{j=1}^{m} f_j(A_j^{\hat{\pi}}) \leq \frac{y_i}{\lambda} \tag{116}$$

Given the second fact, we can prove the third fact as follows. Let $i^* \in \operatorname{argmin}_i y_i$. By definition $x_i = \min\{y_i, c^*\}$, then $i^* \in \operatorname{argmin}_i x_i$. We consider the two cases:

(1) $y_{i^*} \leq c^*$: In this case, we have that $x_{i^*} = y_{i^*}$. Since $x_i \leq y_i, \forall i$, it holds that $\bar{x} \leq \bar{y}$. The third fact follows as $x_{i^*} = y_{i^*} \geq \lambda \bar{y} \geq \lambda \bar{x}$.

(2) $y_{i^*} > c^*$: In this case, $y_i \geq c^*$ holds for all $i$. As a result, we have $x_i = c^*, \forall i$. Therefore, $x_i = \bar{x} = c^* \geq \lambda c^*$.

Combining fact 3 and 4, it follows for each $i$:

$$\bar{\lambda} f_i(A_i^{\hat{\pi}}) + \lambda \frac{1}{m} \sum_{j=1}^{m} f_j(A_j^{\hat{\pi}}) = y_i \geq x_i \geq \lambda \bar{x} \geq \alpha \lambda c^* \geq \alpha \lambda (OPT - \epsilon). \tag{117}$$

The first part of the Theorem is then proved.

The second part of the Theorem simply follows from the proof in Theorem 2.3 and Eqn 117. □

## 6.1 Proof for Theorem 3.3

Define $F^{\lambda}(\pi) = \bar{\lambda} \max_i f_i(A_i^{\pi}) + \lambda \frac{1}{m} \sum_{i=1}^{m} f_i(A_i^{\pi})$ for any $0 \leq \lambda \leq 1$. GENERALLOVÁSZ ROUND is guaranteed to find a partition $\hat{\pi} \in \Pi$ such that

$$F^{\lambda}(\hat{\pi}) \leq m \min_{\pi \in \Pi} F^{\lambda}(\pi) \tag{118}$$

*Proof.* We essentially use the same proof technique in Theorem 2.7 to show this result. After solving for the continuous solution $\{x_i^* \in \mathbb{R}^n\}_{i=1}^{m}$, the rounding step simply chooses for each $j = 1, \ldots, n$, assigns the item to the block $i^* \in \operatorname{argmax}_{i=1,\ldots,m} x_i^*(j)$. We denote the resulting partitioning as $\hat{\pi} = \{A_i^{\hat{\pi}}\}_{i=1}^{m}$.

It suffices to bound the performance loss at the step of rounding the fractional solution $\{x_i^*\}_{i=1}^m$, or equivalently, the following:

$$\tilde{f}_i(x_i^*) \geq \frac{1}{m} f_i(A_i^{\hat{\pi}}), \tag{119}$$

Given Eqn 119, the Theorem follows since

$$F^\lambda(\pi^*) \geq \bar{\lambda} \max_i \tilde{f}_i(x_i^*) + \lambda \frac{1}{m} \sum_{j=1}^m \tilde{f}_j(x_j^*) \tag{120}$$

$$\geq \frac{1}{m} [\bar{\lambda} \max_i f_i(A_i^{\hat{\pi}}) + \frac{\lambda}{m} \sum_{j=1}^m f_j(A_j^{\hat{\pi}})] \tag{121}$$

$$\geq \frac{1}{m} F^\lambda(\hat{\pi}). \tag{122}$$

To prove Eqn 68, consider the following:

$$f_i(A_i^{\hat{\pi}}) = \tilde{f}_i(1_{A_i^{\hat{\pi}}}) = m\tilde{f}_i(\frac{1}{m} 1_{A_i^{\hat{\pi}}}) // \text{ positive homogeneity of Lovász extension} \tag{123}$$

For any item $v_j \in A_i^{\hat{\pi}}$, we have $x_i^*(j) \geq \frac{1}{m}$, since $\sum_{i=1}^m x_i^*(j) \geq 1$ and $x_i^*(j) = \max_{i'} x_{i'}(j)$. Therefore, we have $\frac{1}{m} 1_{A_i} \leq x_i^*$. Since $f_i$ is monotone, its extension $\tilde{f}_i$ is also monotone. As a result, $f_i(A_i) = m\tilde{f}_i(\frac{1}{m} 1_{A_i}) \leq m\tilde{f}_i(x_i^*)$. $\qquad\square$