[Reviews · NeurIPS 2015]

Submitted by Assigned_Reviewer_1

The paper proposes new algorithms to address a set of problems falling under the umbrella term of 'submodular partitioning' -

including two distinct clustering problems, namely clustering

to maximize homogeneity, or clustering so as to maximize the representation power of every cluster (e.g. so as to accelerate distributed learning). The authors consider the case where common costs are applied to each cluster (homogeneous case), or when distinct costs are applied to every cluster (hetergoeneous). Adding to this split, the authors also treat a mix of a robust and an average loss (e.g. a linear combination of the the max distortion with the average distortion when clustering).

Starting from the observation that the algorithms working for the robust variants are not scalable, the authors proceed to propose (i) greedy algorithms for the robust homogeneous/heterogeneous algorithms with approximation guarantees, and (ii) a simple method for blending the solutions to the robust and the average cost optimizations.

They then proceed to practical examples, demonstrating improvements over baseline algorithms for clustering.

Even though the paper is

written by experts who know how to write clearly, I found it really hard to grasp the technical part. In my understanding this is because the authors try to cover too many things (including 6 algorithms, 4 problems, and 7 theorems proved in the appendix). Even though I trust the authors know what they are doing (the appendix provides many more details, but I admit that I do not have the technical background to review them all), I think the paper would benefit from focusing on the single most interesting problem and presenting it more smoothly.

The paper now reads mostly a list of pointers to other works, combinations of either existing results, or results presented in the appendix, and comes with very little explanation of the techniques being used - I was surprised that the Lovasz extension was presented.

I would have

appreciated more reading (and understanding) a paper on one problem, with one algorithm, rather than a problem on 8 problems with 6 algorithms.

I am also not too convinced when it comes to the evaluation of the proposed algorithms. For problems 1-3 the authors only compare to very weak baselines (random splitting). Since in all these three problems the case of lambda = 0 is considered, I would have expected that the authors either compare to the existing algorithms for robust partitioning and report either improvements in time, or in accuracy.

Problem 4 is the sole case where lambda > 0 is used (which is a bit underwhelming, since treating such cases is one of the main contributions of this paper); for that problem (unsupervised segmentation) the authors again compare to weak baselines, not considering e.g. any method that uses graph-cuts. Furthermore, the authors mention in the supplemental material that their method is 'scalable' to this problem while others are not; running times are not reported, while the dependence of their method on image size is not clear. I therefore found it hard to assess how 'scalable' the method actually is.

Suggestions: Apart from the broader suggestion above (to focus on a few aspects of the work), I would propose to (a) avoid covering

prior work as thoroughly, to make up space for your own contributions and (b) maybe move the 'Applications' in page 3 together with the experiments, because after the intervention of 4 pages of theory the reader has already forgotten the motivation.

Minor comments: l. 259: add comma after [17] l. 304: programming -> program
Summary: The paper

considers partitioning problems involving submodular functions and proposes algorithms that are able to scale to large datasets and come with approximation guarantees. I understand that the problem is technically interesting and well-motivated, while the paper's novelty is non-negligible - but due to the paper's density I think its technical contribution will be only accessible to the experts working on the problem.

Submitted by Assigned_Reviewer_2

Summary : The paper studies the following two problems a) Find a partition A1,A2,...An of n elements such that a convex combination of min(fi(Ai)) (robust version) and \sum_{i} fi(Ai) (average case version) is maximized. b) Find a partition A1,A2,...An of n elements such that a convex combination of max(fi(Ai)) (robust version) and \sum_{i} fi(Ai) (average case version) is minimized.

It proposes the following algorithms a) GREEDMAX, GREEDSAT, MMAX for the case of maximizing min(fi(Ai)) b) GREEDMIN, LOVASZROUND, MMIN for the case of minimizing max(fi(Ai)) c) Algorithms for convex combination of robust and average case versions of the problem. Algorithm is essentially solving each of robust and average case versions separately and then taking the best of the two solutions.

The paper further evaluates the GREEDMAX and GREEDMIN algorithms experimentally.

Clarity of the paper: The paper is reasonably well written and understandable.

Significance of the problem: Submodular problems are important class of problems in machine learning as they capture discrete non-linear functions while still allowing tractable algorithms for a large class of problems. Submodular partitioning studied in this paper is on of the harder problems in the class of submodular optimization and has many interesting applications. Scalable algorithms for this problem should be of interest.

Originality: In terms of both modeling and new ideas for algorithms the results in this paper are quite similar to previous papers in

the area.

Quality:

a) While it is true that previous algorithms for these problems have been expensive in terms of running time, variants of greedy are expected to produce solutions whose solution quality can be bounded. It is not very surprising given that the approximation ratio is not necessarily very good.

b) The experimental results are not very convincing. Although the algorithm's convergence rate beats that of random partitioning, the margin seems to be quite small.

c) The algorithm for the mixed robust version (where 0 < lambda < 1) is very unsatisfying as it just picks the best solution of solving robust and average case versions independently. This is specifically the case since we are introducing the model of mixed version as it is supposed to capture something more general than both robust and average case versions.

Summary: The paper considers mixed robust/average submodular data partitioning problems. It proposes and experimentally evaluates several new algorithms such as greedy, majorization-minimization, minorizationmaximization, and relaxation algorithms.

Submitted by Assigned_Reviewer_3

The authors propose study optimization problem of form

max (1 - lambda) * min_i f_i(A_i) + lambda / m sum_j^m f_j(A_j) as well as the minimization version.

They propose several algorithms for special cases lambda = 0, as well as general algorithm for 0 <= lambda <= 1. The algorithm for special cases improve the approximation guarantees of existing state-of-art.

The strongest point of the paper are the two algorithms, GreedMax with 1/m approximation guarantee and LovaszRound with guarantee of m. Both improve the state-of-the-art guarantees. GreedMax seems more practical as LovaszRound requires to solve the convex problem.

The weakest point of the paper is the algorithm for 0 < lambda < 1, due to following reasons: - while the special cases (lambda = 0, 1) do occur in practice, the mixed version doesn't seem

to appear before.

This is fine if authors want to introduce a new type of a

problem, but then it should be better motivated, and experiments should be

more extensive. - The solution for the problem is uninspiring. Essentially, we solve the

problem for lambda = 0 and 1, and select the best one. The algorithm ignores

completely the relationship between the two terms. - This approach actually works for any linear combination,

max alpha * min_i f_i(A_i) + beta / m sum_j^m f_j(A_j)

with alpha, beta >= 0 (naturally one needs to adjust the approximation guarantees)

so expressing the problem as it is done in Eq. 1 reads that the authors were trying

to force this setup.

In the experiments, the authors should compare their optimizers to the baseline

optimizers (listed in Table 1), both in running time as well as quality (objective score).

Other comments:

The paper is extremely crammed and suffers from NIPS style of formatting. I recommend that, at least for the conference version, the authors would reduce the content, by example cutting/reducing the exposure of the generic 0 <= lambda <= 1.

Table 1: at the moment it is overwide. I suggest that you break the tables into two tables: one for Problem 1 and one for Problem 2. Use the additional horizontal space to show running times of each algorithm.

I suggest that you change the partition notation: A^pi_i to A_i and pi to \mathcal{A}

Section 2.1. and Section 2. State in the title (similar to Section 2 title)

which problem and lambda setting you are solving.

The legends and axis in Figure 3 are almost unreadable. Plot your figures such that scaling keeps the ratio aspect constant, as well as enlarge the fonts.
Summary: The paper has interesting algorithms for solving worst-case submodular problems. These algorithms have stronger approximation guarantees than the state-of-the-art. I would like to see some improvements in writing as well as more extensive empirical comparison to the existing algorithms.

Author Feedback
Author rebuttal: We thank all reviewers for their time & valuable comments.

1) Paper too dense (R1)

We to provide a complete & thorough understanding of "submodular partitioning" hence the density (we propose two novel & general forms of submodular partitioning, new analyses, & several new ML applications of these formulations, e.g., data partitioning for distributed ML & image segmentation)). If the paper is accepted, however, we will strive to adjust the density to make it more accessible to readers wishing to read only the 8-page NIPS version.

2) More empirical validation against other data partitioning methods (R3)

Though most previous work on worst-case submodular data partitioning (lambda=0) are theoretically interesting, they are not scalable (lines 84-105). The Ellipsoid algorithm for Problems 1 & 2 takes many hours to run even for small problems (|V| = 60). The sampling-based algorithm involves many instances of submodular minimization, hence is also prohibitive. The matching-based algorithm as well as the balanced algorithm, though slightly more scalable, are not practically interesting since they do not exploit the problem structure. Hence, there are few scalable alternatives.

We believe our empirical results are quite strong, showing the applicability & benefit of our algorithms to a wide variety of new ML applications. However, to make the empirical section still more complete, we will, as you suggest, add more comparison against scalable baselines (we've in fact done this already & they perform worse).

3) Improvements in convergence rate over the random partitioning not significant (R2).

We disagree. We extensively tested our data partitioning method on three different ML data sets for the distributed convex optimization & distributed deep neural network training. In Figure 1-3, we show that the submodular partitioning *consistently* & significantly outperforms the random baseline. It is often observed that submodular partitioning beats the best of 10 instances of random partitioning. Moreover Figure 1-3 suggest that submodular partitioning also often converges to higher performance than that of the random system.

4) Algorithms for the general scenarios (0 < lambda < 1) are weak (R3).

We provide a very simple & generic framework for yielding good theoretical guarantees that interpolate between the two extreme cases (lambda=0 & lambda=1) in terms of lambda. We do not suggest or imply that additional interesting & practically useful algorithms for the general lambda case are impossible, but given we are the first to even consider & offer algorithms for 0 < lambda < 1 having guarantees, & that are practical & scalable as well, we believe our paper offers significant novelty on this front.

5) Lack of motivation & empirical validation for the general scenarios (R1 & R3).

We motivate Problem 2 under the general scenario with the clustering & image segmentation applications (line 120-128). A mix of the worst- & average-case objectives suits better image segmentation problems. We failed to mention a semi-supervised and/or interactive image segmentation case when unary potentials are available, in which case the 0 < \lambda < 1 case is even more relevant (we will add this given the opportunity). We empirically show that significantly better image segmentation results are obtained when formulated in the general scenario (lines 420-430).

6) Both modeling & ideas for algorithms are similar to previous papers in the area (R2).

We disagree. The contributions of this work are quite novel, on both the theoretical & empirical aspects, as explained on lines 154-181. To reiterate (from the paper), this work is the first: to study submodular data partitioning in the general case (0 < lambda < 1); to apply submodular partitioning formulations to image segmentation, distributed convex optimization, & distributed deep NN training, & to show nice empirical results; to propose GREEDMAX, which improves the state-of-the-art theoretical guarantee for the max-min submodular fair allocation problem in the homogeneous case; to propose LovaszRound, which has a guarantee matching the hardness of min-max submodular load balancing problem.

7) Scalability of unsupervised segmentation (R1 & R2).

With n as the number of image pixels, the complexity of unsupervised image segmentation (modified variant of GreedMin) is O(n^2) function valuations. Given the opportunity, we will report run times of our C++ implementations.

8) Lack of comparison against other baseline segmentation methods.

We test unsupervised image segmentation (lines 418-425), inherently a clustering problem. Therefore, we extensively compared against clustering baselines (spectral clustering, k-means, k-medoids). We also have previously tested results for *unsuperivsed* graph-cut based methods but they were very bad and were not considered (we will strive to include them, as requested, if the paper is accepted).